palaeontology/evolution

Teiidae, fossils, apomorphies, Miocene, divergence time, biogeography

**Author for correspondence:**
Simon G. Scarpetta
e-mail: scas100@utexas.edu

# Unusual lizard fossil from the Miocene of Nebraska and a minimum age for cnemidophorine teiids

## Simon G. Scarpetta

Department of Geological Sciences, Jackson School of Geosciences, The University of Texas at Austin, Austin, TX, USA

SGS, 0000-0003-0976-9337

Teiid lizards are well represented in the fossil record and are common components of modern ecosystems in North and South America. Many fossils were referred to the cnemidophorine teiid group (whiptails, racerunners and relatives), particularly from North America. However, systematic interpretations of morphological features in cnemidophorines were hampered by the historically problematic taxonomy of the clade, and the biogeography and chronology of cnemidophorine evolution in North America is poorly understood from the fossil record. Few fossil cnemidophorines were identified with an apomorphy-based diagnosis, and there are almost no fossil cnemidophorines that could be used to anchor node calibrations. Here, I describe a cnemidophorine from the Miocene Ogallala Group of Nebraska and diagnose the fossil using apomorphies. In that process, I clarify the systematic utility of several morphological features of cnemidophorine lizards. I refer the fossil to the least inclusive clade containing *Aspidoscelis*, *Holcosus* and *Pholidoscelis*. The most conservative minimum age of the locality of the fossil is a fission-track date of 6.3 Ma, but mammal biochronology provides a more refined age of 9.4 Ma, which can be used as a minimum age for the crown cnemidophorine clade in divergence time analyses. The fossil indicates that a cnemidophorine lineage that does not live in Nebraska today inhabited the area during the Miocene. I refrain from naming a new taxon pending discovery of additional fossil material of the lineage to which the fossil belonged.

## 1. Introduction

Teiidae is a clade of diurnal and largely terrestrial lizards with a substantial Cenozoic fossil record from North and South America [1–3]. Borioteiioidea, a clade hypothesized to be the sister group of extant Teiidae ([4,5], but see [6–10]), is known

**Figure 1.** Map of published fossil localities from the Neogene located in continental North America (north of Mexico) that have reported fossil teiids. The base map was created in RStudio [31] using the package *tmap* [32]. The locality from the present study (4) is marked with a circle. Stars mark other localities. Locality colours denote geologic age, and numbers correspond to records from the following publications: **1** [26]; **2** [28]; **3** [27]; **4** [23] and present study; **5** [30]; **6** [25]; **7** [24]; **8** [29]; **9** [20]; **10** [21]; **11** [22].

from the Cretaceous of North America and Eurasia. Barbatteiidae, another potential sister clade of extant Teiidae, is known from the Cretaceous of Europe [11,12].

Extant Teiidae is divided into two clades, a clade containing the large-bodied tegus and their relatives (Tupinambinae), and Teiinae, a clade containing *Dicrodon*, *Teius* and a group known as the cnemidophorines (whiptails, racerunners and relatives). Extant cnemidophorines are widespread and common components of modern North American ecosystems [3]. Fossil cnemidophorines were recovered in large numbers from Quaternary deposits on mainland North America (north of Mexico) [13–15] and the Caribbean [16–19]. Some fossil cnemidophorines were reported from mainland Neogene localities [20–30] (figure 1). There are no cnemidophorines or other teiids known from the Palaeogene of North America.

Most fossil cnemidophorines from the Neogene of North America are isolated and fragmented dentaries and maxillae. The identifications of those fossils were hindered by problematic taxonomy of cnemidophorines [13,22,26], and few of those fossils were diagnosed with apomorphies. Divergence time analyses indicate that most crown cnemidophorines originated during the Oligocene and Miocene [33,34], and many teiid skeletal elements are distinct from those of other squamates [13,35] and so should be diagnosable through an apomorphy-based approach [36]. Thus, the paucity of unambiguously identified fossil cnemidophorines from the Neogene of North America is striking, as is the absence of cnemidophorines from the late Palaeogene. The evolutionary history of cnemidophorines is not well understood from the Neogene fossil record of North America.

I describe a nearly complete and unusual fossil dentary of a cnemidophorine lizard from the Miocene Ogallala Group of Cherry County, Nebraska. The fossil is a marginal tooth-bearing element like most other known teiid fossils from the Neogene, but it is largely complete and preserves morphologies that allow for an apomorphy-based referral to the crown cnemidophorine clade. The fossil provides new information on the temporal and biogeographic history of cnemidophorines in North America. A single fossil referred to cf. *Cnemidophorus* was previously reported from Cherry County in the Valentine Formation (figure 1), but the fossil was lost before publication [23].

# 2. Material and methods

## 2.1. Age, geologic setting and collection of specimen

The specimen is reposited at the Yale Peabody Museum and was collected by Oscar Harger on the Yale College scientific expedition of 1873. Unfortunately, the available locality information is limited to the Ogallala Group at the Niobrara River north of Minnechaduza Creek in Cherry County, Nebraska. The Ogallala Group in that region of Cherry County is represented by the Valentine and Ash Hollow

formations [37]. Thus, the maximum age of the fossil is $13.5 \pm 0.01$ Ma based on interpolation of the age of the Hurlbut Ash between dated horizons [38] or $13.55 \pm 0.09$ Ma based on $^{40}$Ar/$^{39}$Ar dates for the Hurlbut Ash [39], and the minimum age is $6.6 \pm 0.3$ Ma based on fission-track of glass from near the top of the Ash Hollow Formation [37,40].

The age of the fossil is further constrained by biochronology of North American mammals in the Valentine and Ash Hollow formations. Although there are no mammals that can be attributed to the exact site where YPM VP 4707 was collected, fossil mammals from other known localities in the Valentine and Ash Hollow formations in Cherry County are characteristic of the Barstovian and Clarendonian North American land mammal ages (NALMAs) [41]. The Barstovian precedes the Clarendonian. There are no documented localities from the Ogallala Group anywhere in Cherry County that contain taxa characteristic of the younger Hemphillian NALMA, which follows the Clarendonian and is considered to extend to the early Pliocene [40,41]. Key taxa from Clarendonian localities in Cherry County include *Eubelodon* (a gomphothere) and *Barbourofelis* (a feliform carnivoran) [41]. The Clarendonian NALMA is currently considered to extend to 9.4 Ma [42]. That age could change if the dates bracketing the extinction of taxa at the end of the Clarendonian (e.g. *Eucastor, Aelurodon*) are revised, or if specimens of those taxa are found that post-date 9.4 Ma [40,42]. Conservatively, the minimum age of the fossil is 6.3 Ma, but based on mammal biochronology, a minimum age of 9.4 Ma is assigned.

## 2.2. Institutional abbreviations

CAS, California Academy of Sciences, San Francisco, CA; MVZ, Museum of Vertebrate Zoology, Herpetology Collection, University of California, Berkeley, CA; TNHC, Biodiversity Collections, Herpetology Collections (Texas Natural History Collections), The University of Texas at Austin, TX; TxVP, Texas Vertebrate Paleontology, The University of Texas at Austin, TX (formerly TMM); UF, Florida Museum of Natural History, Herpetology Division, University of Florida, Gainesville, FL; YPM, Yale Peabody Museum, New Haven, CT.

## 2.3. Terminology, taxonomy and specimens examined

Osteological terminology follows Evans [43] unless otherwise noted. Taxonomy of cnemidophorines follows Tucker *et al.* [33,44]. I accept the phylogenetic hypotheses of Tucker *et al.* [33,44] in my diagnoses, most importantly, that *Aspidoscelis*, *Holcosus* and *Pholidoscelis* form a clade to the exclusion of other extant teiid lizards, and that *Aspidoscelis* and *Holcosus* are sister taxa. A complete list of specimens examined is in electronic supplementary material, file S1. I examined at least one specimen of all cnemidophorines besides *Contomastix*. I also examined *Callopistes* and all tupinambines besides *Crocodilurus*.

Comparative specimens of the cnemidophorines *Ameivula* [45] and *Aurivela* [46] are based upon computed-tomography scans. The data were accessed from http://www.morphosource.org/Detail/MediaDetail/Show/media_id/45303 and http://www.morphosource.org/Detail/MediaDetail/Show/media_id/45295. I did not perform segmentation of the scans and examined the specimens as volume renderings in Avizo Lite 2019.

# 3. Results

## 3.1. Systematic Palaeontology

Teiidae Gray 1827 [47]
*Aspidoscelis* Fitzinger 1843 [48]; *Holcosus* Cope 1862 [49]; *Pholidoscelis* Fitzinger 1843 [48]
Unnamed clade containing *Aspidoscelis*, *Holcosus* and *Pholidoscelis* sp.
Referred specimen: YPM VP 4707
Figure 2.

## 3.2. Description

YPM VP 4707 is a mostly complete right dentary that preserves the tooth row, four teeth, the symphysis and some of the coronoid and angular processes (figure 2*a,b*). The posterior portion of the dentary is

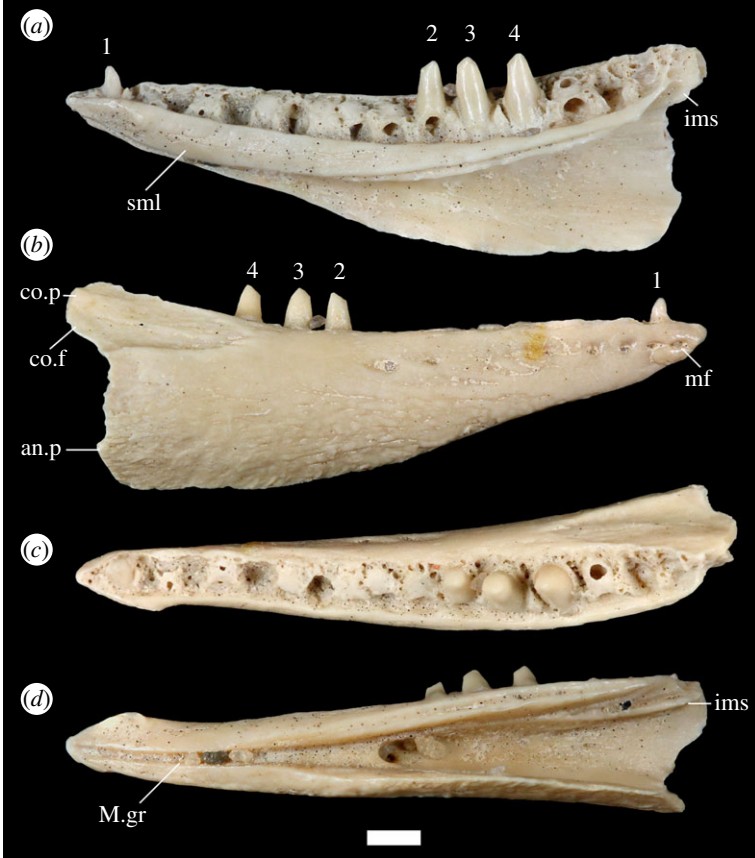

**Figure 2.** Fossil teiid YPM VP 4707. Tooth numbers (referenced in the text) are indicated above each tooth in (*a*) and (*b*). Scale bar equals 1 mm. (*a*) Lingual view; (*b*) labial view; (*c*) dorsal view; and (*d*) ventral view. an.p, angular process; co.f, coronoid facet; co.p, coronoid process; ims, intramandibular septum; mf, mental foramen; M.gr, Meckelian groove; sml, suprameckelian lip.

significantly taller than the anterior portion of the dentary, and the dentary dramatically tapers in height anteriorly. The Meckelian groove is long and open along the entire dentary. Anteriorly, the groove faces ventrally and is restricted by the suprameckelian lip (*sensu* [50]). The posterior portion of the groove has a deep mediolateral dimension, at least in part due to a lateral expansion of the dentary at the coronoid facet (figure 2*c*). The Meckelian groove leaves space for the splenial to extend far anteriorly, almost to the symphysial facet (figure 2*a*,*d*). The suprameckelian lip has a relatively short dorsoventral dimension posteriorly, but markedly increases in height anteriorly. There is an inset articulation facet for the splenial along the ventral margin of the posterior half of the suprameckelian lip. Lateral to the suprameckelian lip, there is a posterior exposure of the intramandibular septum (the intramandibular lamella; [51,52]), which would separate the anteromedial process of the coronoid from the surangular. The intramandibular septum is dorsoventrally short, especially anteriorly (figure 2*d*). The posterior portion of the septum reaches anteriorly to the fourth tooth (numbered on figure 2*a*,*b*), and the anterior portion, a slight ridge in ventral view, reaches the second tooth. The subdental gutter is relatively deep anteriorly and shallower posteriorly. Some of the tooth replacement pits invade the subdental gutter (figure 2*c*). The subdental gutter extends across the entire tooth row and has a moderately broad lingual dimension, although it narrows anteriorly near the symphysis.

The tooth row has a length of 11.37 mm. There are 19 apparent tooth positions, but it is difficult to distinguish the mesialmost tooth positions, so that count should be viewed as tentative. Three distal teeth and one mesial tooth are preserved. The bases of the three distal teeth are adjacent but do not contact. Dentition is pleurodont and is heterodont in terms of both tooth size and cusp morphology. The distalmost tooth has a mesiodistally expanded base, and the two teeth mesial to that tooth taper less distinctly from the base to the crown. The crowns of the three distal teeth are asymmetrically and mesiodistally bicuspid, and the mesial crown is smaller than the main crown. The main (distal) crown of the third tooth slopes ventrodistally instead of ventrally to create a slight distal shoulder. The mesial crowns of all three biscuspid teeth are apically worn. The mesial tooth is unicuspid and is substantially smaller than the distal teeth. A replacement pit is present on the second tooth, and pits

are present within dental tissues at other tooth positions that lack teeth. The replacement pits are deep and almost circular. There are substantial but not excessive deposits of basal cementum at the bases of the teeth and at tooth positions lacking teeth, and the cementum does not fill the subdental gutter.

The labial face of the dentary is moderately convex and has rugose sculpturing across much of its surface, but rugosities are absent immediately ventral to the three distal teeth and on the anteriormost portion of the bone (figure 2b). The coronoid facet is subtriangular and extends anteriorly to the fourth tooth, and ventrally to the ventrolateral extent of the posterolateral expansion of the dentary. The facet is deep, moderately textured, and has a marked ventral boundary. There are nine distinct nutrient (mental) foramina that extend posteriorly to the second tooth, although most of the foramina are concentrated anteriorly. The posterior portion of the dentary is less complete than the rest of the fossil, and so the posterior extent of the coronoid and angular processes is uncertain, as is the presence of a separate surangular process.

## 3.3. Diagnosis

YPM VP 4707 is referred to the least inclusive clade containing *Aspidoscelis*, *Holcosus* and *Pholidoscelis* based on the presence or absence of the following morphological features; the clade for which the feature is hypothesized to be an apomorphy is listed in parentheses: pleurodont tooth implantation (Lepidosauria), substantial deposits of basal cementum (Teiidae), an elongate Meckelian groove providing space for a hypertrophied splenial (Teiidae), large subcircular tooth replacement pits (Teiidae), an open Meckelian groove (Teiidae), Meckelian groove restricted anteriorly (Teiinae), the absence of a large incision between the coronoid and surangular processes (Teiinae), sculpturing on the labial surface of the dentary both anteriorly and posteriorly (least inclusive clade containing *Aspidoscelis*, *Holcosus* and *Pholidoscelis*), dentary tapers in height posteriorly to anteriorly (present in several cnemidophorines, see below), and bicuspid distal teeth (present in several cnemidophorines, see below).

## 3.4. Expanded diagnosis and discussion

The presence of pleurodont tooth implantation (see [53]) diagnoses YPM VP 4707 as a member of Lepidosauria [5,35,54]. Rhyncocephalians generally have acrodont tooth implantation and apically positioned teeth, although *Gephyrosaurus* has more apically positioned teeth that are considered pleurodont [5,55].

Asymmetrically bicuspid distal teeth and an open Meckelian groove are present in Teiidae, but are also present in Lacertidae, which is closely related to Teiidae [34]. Gymnophthalmidae+Alopoglossidae, the sister clade of Teiidae among extant squamates [34], can also have bicuspid teeth, but the Meckelian groove is generally closed and fused [56,57]. Amphisbaenia, the sister taxon of Lacertidae, possesses several unique derived features, including unicuspid teeth, a short dentary, a low tooth count (less than 10 teeth), and dentition described as acrodont, subacrodont or subpleurodont [58,59]; the dentition of teiids has also been described as subpleurodont [60]. YPM VP 4707 has substantial deposits of cementum at its tooth bases, large subcircular tooth replacement pits, and an elongate Meckelian groove providing space for a hypertrophied splenial, all three of which are apomorphies of Teiidae to the exclusion of other squamates ([4,35]; see below).

Borioteiioids can have all three dentary apomorphies commonly used to diagnose teiid fossils [4] that are present in YPM VP 4707. The fossil is excluded from Polyglyphanodontia because it lacks distal teeth with transversely oriented cusps, from Chamopsiidae because it lacks a marked symphysial boss and possesses an intramandibular septum [61], from *Prototeius* because the distal teeth are bicuspid and are not massive or blunt-cusped and the Meckelian groove is anteriorly restricted [51], and from members of Macrocephalosauridae, which have flared, multicuspid teeth, sometimes with obliquely oriented cusps [62].

Members of Barbatteiidae were also reported to possess the three dentary apomorphies of Teiidae, and the clade was hypothesized to be the sister group of Teiidae *sensu stricto* [11,12]. YPM VP 4707 is differentiated from documented dentaries of barbatteiids because it tapers in height anteriorly, it lacks an incision between the coronoid and angular processes, and it possesses mesiodistally expanded distal tooth bases, labial sculpturing, and a better-developed subdental gutter in terms of both dorsoventral depth and lingual width [12]. YPM VP 4707 is part of total clade Teiidae *sensu stricto* (excluding Borioteiioidea and Barbatteiidae).

The anterior restriction of the Meckelian groove excludes the fossil from Tupinambinae and *Callopistes*, whose members have a completely unrestricted and medially facing Meckelian groove

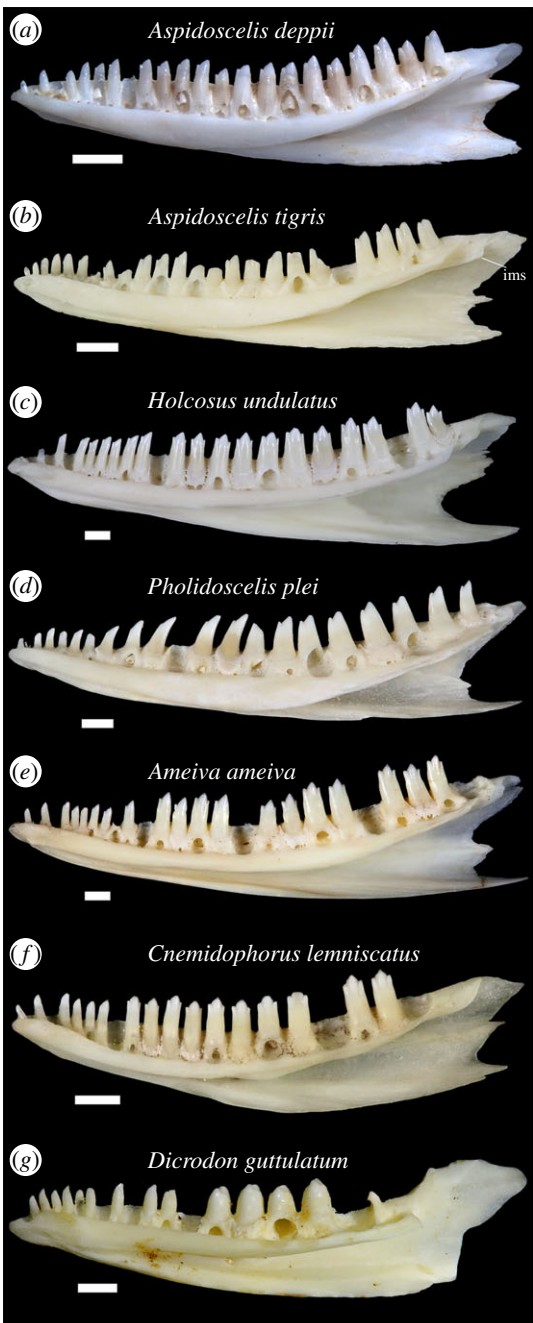

**Figure 3.** Right dentaries of extant teiids in lingual view. Scale bars equal 1 mm. (*a*) *Aspidoscelis deppii* TNHC 96795; (*b*) *Aspidoscelis tigris* TxVP 8629; (*c*) *Holcosus undulatus* UF 51244; (*d*) *Pholidoscelis plei* UF 22260; (*e*) *Ameiva ameiva* UF 57896; (*f*) *Cnemidophorus lemniscatus* UF 76231; and (*g*) *Dicrodon guttulatum* MVZ 77474. ims, intramandibular septum (posterior portion is the intramandibular lamella).

[51,63], but is a feature shared with Teiinae (figure 3). Tupinambines and *Callopistes* also differ from the fossil by possessing a large incision in between the coronoid and angular processes of the dentary. Tupinambines and *Callopistes* are further distinguished from YPM VP 4707 because they have more extensive deposits of basal cementum, which fills the subdental gutter and creates mesial and distal walls between teeth (subthecodont implanation) [53,63]. The teiines *Dicrodon* and *Teius* also have considerably more basal cementum relative to cnemidophorines and YPM VP 4707 (Fig 3G; [63]). The fossil lacks the transverse tooth cusps of *Dicrodon* (figure 5*g*) and *Teius* [60,64,65].

Labial sculpturing was previously reported to occur in some adult individuals of *Ameiva* and *Cnemidophorus* without reference to any particular species [26] and before those genera were split up. Among examined extant teiids (see electronic supplementary material, file S1), only *Aspidoscelis* and *Holcosus* have sculpturing on the labial surface of the dentary (figure 4*a*–*c*). Sculpturing on

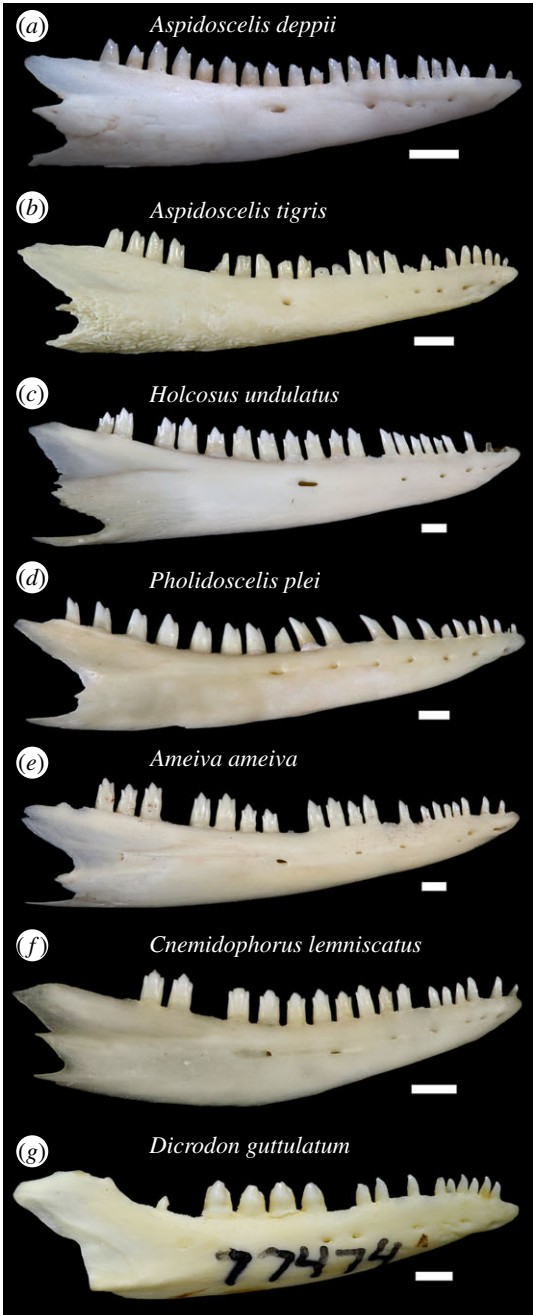

**Figure 4.** Right dentaries of extant teiids in labial view. Scale bars equal 1 mm. (*a*) *Aspidoscelis deppii* TNHC 96795; (*b*) *Aspidoscelis tigris* TxVP 8629; (*c*) *Holcosus undulatus* UF 51244; (*d*) *Pholidoscelis plei* UF 22260; (*e*) *Ameiva ameiva* UF 57896; (*f*) *Cnemidophorus lemniscatus* UF 76231; and (*g*) *Dicrodon guttulatum* MVZ 77474.

the anterolabial surface of the bone was previously documented in *Pholidoscelis* [17]. I interpret labial sculpturing as an apomorphy of the least inclusive clade containing *Aspidoscelis*, *Holcosus* and *Pholidoscelis*.

Most specimens of *Aspidoscelis* lack labial sculpturing. In both *Aspidoscelis* and *Holcosus*, sculpturing occurs often, but not universally, on larger and more robust specimens that were probably skeletally mature adults, as was observed by Norell [26]. *Holcosus festivus* MVZ 79608 is a larger specimen that lacks sculpturing. Sculpturing is present on all examined specimens of *Aspidoscelis deppii*, some specimens of *Aspidoscelis tigris* (e.g. TxVP M-8629, M-8631, M-15034), *Aspidoscelis gularis* TxVP M-15028, and on *Holcosus quadrilineatus* UF 37170 and *Holcosus undulatus* UF 51244. In *Holcosus*, labial sculpturing ranges from longitudinal and wispy texturing (*H. undulatus*; figure 4*c*) to more pronounced and vermiculate sculpturing (*H. quadrilineatus* UF 37170). In *Aspidoscelis*, labial

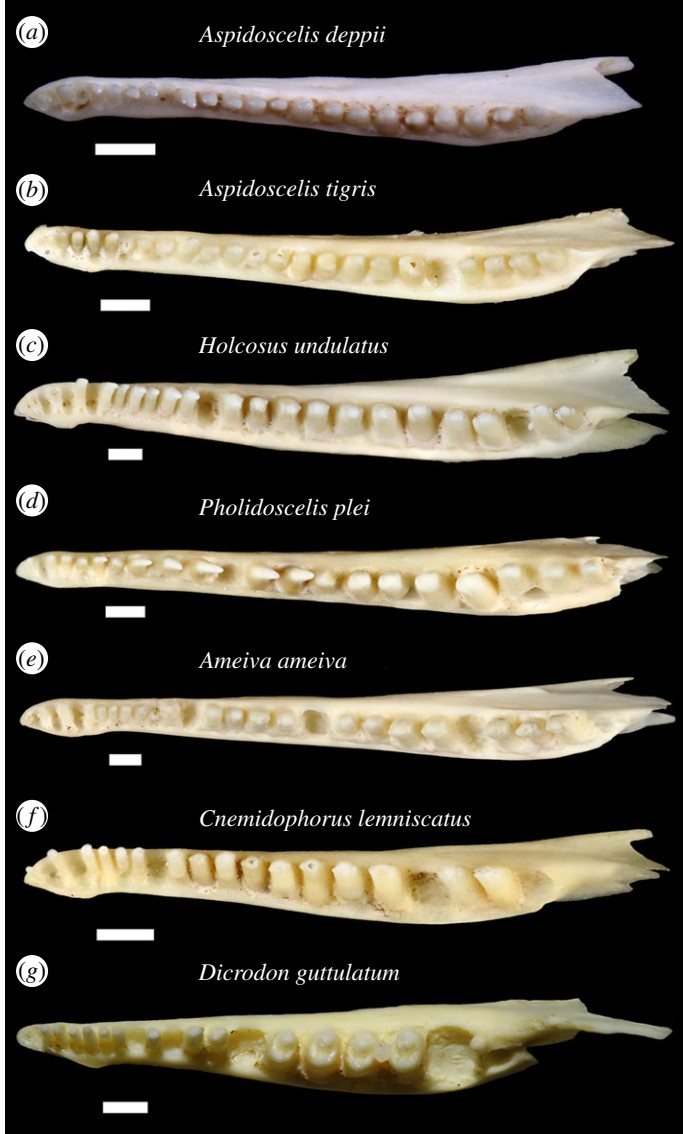

**Figure 5.** Right dentaries of extant teiids in dorsal view. Scale bars equal 1 mm. (*a*) *Aspidoscelis deppii* TNHC 96795; (*b*) *Aspidoscelis tigris* TxVP 8629; (*c*) *Holcosus undulatus* UF 51244; (*d*) *Pholidoscelis plei* UF 22260; (*e*) *Ameiva ameiva* UF 57896; (*f*) *Cnemidophorus lemniscatus* UF 76231; and (*g*) *Dicrodon guttulatum* MVZ 77474.

sculpturing can be rugose (*A. tigris*; figure 4*b*) or less complexly textured (*A. deppii*; figure 4*a*). The sculpturing of YPM VP 4707 is most similar in its rugose texture and its broad extent to examined specimens of *A. tigris*. The unusually complex phylogeny of *Aspidoscelis* [66,67] and intraspecific and intra-clade variability in dentary sculpturing makes the evolution of that morphology difficult to interpret within *Aspidoscelis*. Some specimens of *A. tigris* have highly reduced sculpturing, including larger specimens (e.g. TxVP M-8630). *Aspidoscelis hyperythrus*, a smaller species which may be closely related to *A. deppii* [67], lacks sculpturing altogether.

Tooth cusp morphology is known to vary ontogenetically and interspecifically in teiids [68]. All examined specimens of *Holcosus*, *Ameiva*, *Cnemidophorus*, *Kentropyx* and *Medopheos* have exclusively tricuspid distal dentary teeth, and most have tricuspid median teeth as well. *Aurivela*, *Contomastix* and *Ameivula* were also reported to have tricuspid distal teeth [22,60,69], although the one specimen of *Aurivela* examined here has bicuspid distal teeth. At least one large and presumably adult specimen of *Ameiva* is known to have bicuspid distal teeth [17]. In *Holcosus*, *Ameiva*, *Cnemidophorus*, *Kentropyx* and *Medopheos*, the transition from unicuspid teeth to tricuspid teeth occurs in the anterior portion of the tooth row, often abruptly. Many *Aspidoscelis* have bicuspid teeth throughout the tooth row, but some species, such as *A. uniparens* and *A. sonorae*, have mostly tricuspid teeth.

Intraspecific and ontogenetic variation in tooth morphology is particularly well documented in *Pholidoscelis* [17]. Small specimens have unicuspid mesial teeth, bicuspid median teeth and bicuspid or tricuspid distal teeth, while in larger specimens the teeth generally lose their cusps and become bicuspid or unicuspid, including the distal teeth [17–19]. Some specimens progress to have molarized crushing teeth [17,19]. My observations agree with those of previous studies. Some smaller *Pholidoscelis* that I examined had mostly tricuspid distal teeth (e.g. *Pholidoscelis chrysolaemus* UF 99352), but in larger specimens, distal teeth were all bicuspid (figures 3*d* and 4*d*) or are mostly bicuspid except for the teeth at the last two or three tooth positions, which were tricuspid (e.g. *Pholidoscelis chrysolaemus* UF 99646).

The third tooth of YPM VP 4707 is sloped ventrodistally to create a slight dorsal shoulder on the distal face of the tooth. That morphology also occurs in some specimens of *Aspidoscelis* that have bicuspid distal teeth (e.g. *A. deppii*, figures 3*a* and 4*a*). I interpret the third tooth of YPM VP 4707 as representing a shoulder instead of a third cusp, but recognize that there is some ambiguity. Additionally, the three distalmost tooth positions of YPM VP 4707 are not preserved and could have been tricuspid.

The presence of bicuspid teeth in the distal portion of the tooth row of YPM VP 4707 is most consistent with either *Aspidoscelis* or *Pholidoscelis*. The propensity to have bicuspid distal teeth in adult specimens is probably a derived feature of the clade containing *Aspidoscelis*, *Holcosus* and *Pholidoscelis*, with an apparent reversal in *Holcosus*. However, that feature cannot be used by itself to identify fossil *Aspidoscelis* and *Pholidoscelis*, because bicuspid distal teeth occur in at least a few specimens of other cnemidophorines.

In all examined teiids, the posterior portion of the dentary is taller than the anterior portion, but YPM VP 4707 is unusual in that the anterior portion of the dentary strongly and abruptly tapers in height. I observed more moderate tapering of the dentary in *C. lemniscatus* (figures 3*f* and 4*f*), *A. tigris* (e.g. TxVP M-15034, M-15035) and *H. undulatus* (figures 3*c* and 4*c*). The anteriormost portion of the dentary tapers in *D. guttulatum* (figures 3*g* and 4*g*). Tapering was also previously reported in *Pholidoscelis* [17,18], and strong tapering more comparable with the fossil was illustrated for one large specimen of *P. griswoldi* [19]. A tapering dentary appears to at least be derived in Teiinae; the anterior height of the dentary decreases uniformly in Tupinambinae and *Callopistes*.

Posterolateral expansion of the dentary is most pronounced in *Holcosus* among extant teiids (figure 5*c*). Correspondingly, the ventral border of the coronoid facet is most distinct in *Holcosus*. The facet has a less defined ventral border in *Aspidoscelis*, *Pholidoscelis* and other cnemidophorines, and expansion of the dentary is less exaggerated in those taxa as well. In all cnemidophorines, lateral expansion is most exaggerated in larger specimens. Expansion of the dentary is comparable between YPM VP 4707 and larger specimens of *Holcosus*.

The presence of a posterior extension of the intramandibular septum (the intramandibular lamella) was reported to diagnose polyglyphanodontids to the exclusion of extant teiids [51]. I observed that morphology in several extant teiids, including *Aspidoscelis* and *Holcosus* (figure 3*a–c*).

# 4. Discussion

To my knowledge, YPM VP 4707 is the oldest published record of a crown cnemidophorine from North America. The fossil was identified using apomorphies grounded in a phylogenetic hypothesis based on analyses of targeted-sequence capture data [33,44]. The minimum age of the total clade containing *Aspidoscelis*, *Holcosus* and *Pholidoscelis* is at least 6.3 Ma given a fission-track age estimate of glass from the Ash Hollow Formation [37], but is 9.4 Ma based on the currently accepted temporal extent of the Clarendonian NALMA. The fossil can be used in divergence time analyses as a minimum age of crown cnemidophorines and could be further used to calibrate the minimum age of the split between the clade containing *Aspidoscelis*, *Holcosus* and *Pholidoscelis*, and the clade containing *Ameiva*, *Cnemidophorus*, *Kentropyx* and *Medopheos*. Few fossils were previously considered reliable for bracketing minimum clade ages in divergence time analyses of Teiidae, and only one cnemidophorine of indeterminate phylogenetic affinity was used recently [33,63]. YPM VP 4707 is useful for future studies seeking to temporally calibrate the evolutionary history of teiids and cnemidophorines in particular.

The resurrection of the genera *Aspidoscelis*, *Holcosus* and *Pholidoscelis* for clades previously accommodated in *Cnemidophorus* and *Ameiva* [67,69,70] and the clarification of the phylogeny of Teiinae [33,44] helped elucidate the systematic significance of two morphologies in cnemidophorines. Labial sculpturing of the dentary was reported in adult specimens of *Ameiva* and *Cnemidophorus* by Norell [26]. Sculpturing is restricted to three clades formerly in *Ameiva* and *Cnemidophorus*, the genera

*Holcosus*, *Pholidoscelis* and *Aspidoscelis*. Labial dentary sculpturing is hypothesized to be an apomorphy of the clade containing those three genera. Similarly, bicuspid distal tooth crowns were previously reported in certain groups of *Ameiva* and *Cnemidophorus* [22,60], but *Pholidoscelis* and *Aspidoscelis* are the only genera previously assigned to *Ameiva* and *Cnemidophorus* that consistently have bicuspid distal teeth. The presence of bicuspid tooth crowns cannot be used by itself to identify *Aspidoscelis* or *Pholidoscelis*, however, because bicuspid distal teeth are present in rare specimens of *Ameiva* [17] and *Aurivela*.

Fossil cnemidophorines reported from the early Miocene of Florida [21,22] are consistent with divergence time hypotheses indicating a Miocene or Oligocene origin of crown cnemidophorine clades [33]. However, those fossils were not [22] or could not [21] be referred to the crown clade with an apomorphy-based approach, and so the early evolution of crown cnemidophorines has been poorly understood from the fossil record. YPM VP 4707 confirms the presence of crown cnemidophorines in the middle-late Miocene and indicates that the entire clade occupied at least part of its modern range in North America at that time. It is noteworthy that the fossil is a part of the clade containing *Aspidoscelis*, *Holcosus* and *Pholidoscelis*. Those three genera either inhabit Nebraska in the modern biota (*Aspidoscelis*) or are geographically more proximate to the area than are other cnemidophorines (e.g. *Kentropyx*, *Ameiva*, *Ameivula*) that are now mostly found in South America.

*Aspidoscelis sexlineatus* is the only teiid that occurs in Nebraska or immediately adjacent to Nebraska in the modern biota [71]. Because YPM VP 4707 is not clearly part of crown *Aspidoscelis*, much less the clade containing *A. sexlineatus*, it appears that a different lineage of crown cnemidophorine was found in the area at least as recently as the middle-late Miocene. *Aspidoscelis tigris*, the *A. tesselatus* complex, the *A. neotesselatus* complex and *Aspidoscelis velox* are currently found in southern and western Colorado [71]. *Pholidoscelis* is currently restricted to islands in the Caribbean, and *Holcosus* is found in Central and South America and in Mexico as far north as central Tamaulipas [70,72]. More fossils from across North America are needed to explore the historical biogeography of cnemidophorine lizards and to determine whether modern tropical or island clades like *Holcosus* and *Pholidoscelis* were once found farther north on the mainland or on the mainland at all, respectively.

The present survey of extant teiids indicates that the overall morphotype of YPM VP 4707 is distinctive. However, the fossil cannot be definitively excluded from total or crown *Aspidoscelis*, *Pholidoscelis* or *Holcosus*, so for now I refrain from establishing a new taxon; new fossil material may reveal that YPM VP 4707 is referable to one of those clades. Additional fossils of the lineage to which YPM VP 4707 belonged are needed to refine its systematic position, to determine whether a new taxon is warranted, and to further investigate the biogeographic history of cnemidophorine lizards during the Neogene in North America.

Ethics. YPM VP 4707 is reposited at the Yale Peabody Museum.

Data accessibility. Electronic supplementary material is available in S1.

Competing interests. The author declares no competing interests.

Funding. Funding was provided by a Lundelius award in vertebrate paleontology from the Jackson School of Geosciences at the University of Texas at Austin.

Acknowledgements. I thank Carol Spencer (MVZ), Travis LaDuc (TNHC), Chris Sagebiel (TxVP), Coleman Sheehy III (UF) and Dan Brinkman (YPM) for access to skeletal and fossil material. Chris Bell provided helpful comments on the manuscript and access to literature. Corentin Bochaton, Santiago Brizuela, an anonymous reviewer, and the editor Emily Lindsey provided constructive and thoughtful comments that greatly improved the manuscript. The collection of the CT data for *Ameivula ocellifera* CAS 49378 and *Aurivela longicauda* CAS 18414 was funded by oVert TCN (NSF DBI-1701714, NSF DBI-1701713, NSF DBI-1701870) and the datasets were downloaded from MorphoSource.org.

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
