## [Reviewer comments · Royal Society Open Science]

Review History

RSOS-200317.R0 (Original submission)

Review form: Reviewer 1 (Corentin Bochaton)

Is the manuscript scientifically sound in its present form?

No

Are the interpretations and conclusions justified by the results?

Yes

Is the language acceptable?

Yes

Do you have any ethical concerns with this paper?

No

Have you any concerns about statistical analyses in this paper?

No

Recommendation?

Major revision is needed (please make suggestions in comments)

Comments to the Author(s)

Dear Author,

Your paper constitutes, in my opinion, an interesting discovery that will for sure help to better understand the history of this poorly known (from a paleontological and osteological point of view) group of lizards.

I am not a specialist of North American squamates so I will not be able to provide much advice regarding most of the taxa discussed in your paper but I however did some work on the osteology of *Pholidoscelis*. As such, I believe that more precision could easily be added in the paper regarding the dentary and teeth morphology of this genus. All my comments regarding this are based on a work recently published regarding the fossil *Pholidoscelis* of Guadeloupe in the framework of which I wrote a short synthesis regarding the teeth morphology and osteological variability of modern *Pholidoscelis*. You seem to have missed this publication (Bochaton et al., 2019) which could maybe help you to bring more precision to the following points:

-I think your description of the ontogenetic evolution of the teeth of *Pholidoscelis* lacks precision and reference to the appropriate literature. This variability was first described by Pregill (Pregill, 1981; Pregill et al., 1988) and I tried to expand the description with specimens of most of the species (Bochaton et al., 2019). The general ontogenetic morphological evolution regarding the posterior teeth is tricuspid->bicuspid->monocuspid->bulbous in Lesser Antillean species. In the Greater Antilles most species "stop" this evolution before reaching the last steps and preserve tricuspid or bicuspid posterior teeth in large/adult specimens. This is however not a strict rule (see my comment on *Ameiva*).

-*Pholidoscelis* dentaries of large specimens do present some dermal ornamentation. Not as on your fossil through but limited to the anterior part of the bone (Bochaton et al., 2019).

-Some large specimens of *Ameiva* present bicuspid posterior/distal teeth. This was observed on a specimen from the MNHN, Paris (Bochaton et al., 2019).

-The number of teeth of your specimen (19) falls into the variability of *Pholidoscelis* (15 to 27) (Bochaton et al., 2019). A comparison between the length and the dental row (absent in the paper?) and the number of teeth would maybe help to discard an attribution to *Pholidoscelis*? These comments led to the conclusion that the fossil described in this study cannot be attributed to *Pholidoscelis*, considering that the pattern of dermal sculpturing present on the fossil was previously stated in the literature as absent in this genus. When I first saw the fossil I have been impressed by the posterior height of its labial margin which is also high in large *Pholidoscelis* (there is ontogenetic variability in this). But your fossil seems very small (10.5mm of dental length following the picture) compared to the member of *Pholidoscelis* in which a similar morphology occurs (20-30mm of dental row length).

Another very important point regarding your interpretation of the fossil: in regard to lesser Antillean "*Ameiva*" (= *Pholidoscelis*), Hedges (2006) indicates a divergence time of 45Ma, so much older than your fossil. This should be discussed as it is in strong contradiction with your interpretation.

Here are also other comments regarding the overall presentation of the study:

Introduction

The introduction seems incomplete and is mostly limited to contextual information. The question of the study is not clearly stated here. What is the aim of publishing this fossil? Why was it studied? The introduction contains sufficient context regarding other finds of fossil teiids in the Americas but this part would be much better with a broader introduction at the start and a proper introduction of the research questions.

Also regarding your references, could you split them regarding the different geographic areas, even in a single sentence? This would make easier to know which author speaks about what.

Geological context

This part also lacks important information in my opinion. That would be great to see at least a rapid presentation of the other taxa found in that formation, squamates, or not. That would also help to better understand why this fossil is important. I discovered in the caption of figure 1 that

this locality was previously studied as there is another reference for it? This should be clearly presented in this part. A reference to Fig. 1 should also be added.

Also, you keep referring to the fact that more material would be needed to identify this fossil but it is unclear whether this material exists/could exist or not. Could you clarify this?

Description

Please add the usual measurements in the description, measuring it on the picture might be a source of error.

I hope you will find these few remarks useful to improve this already very interesting study,

Sincerely,

Corentin Bochaton

References

Bochaton, C., R. Boistel, S. Grouard, I. Ineich, A. Tresset, and S. Bailon. 2019. Evolution, diversity and interactions with past human populations of recently extinct *Pholidoscelis* lizards (Squamata: Teiidae) from the Guadeloupe Islands (French West-Indies). *Historical Biology* 31:140–156.

Pregill, G. K. 1981. Late Pleistocene herpetofaunas from Puerto Rico. University of Kansas Museum of Natural History Miscellaneous Publication 71:1–72.

Pregill, G. K., D. W. Steadman, S. L. Olson, and F. V. Grady. 1988. Late Holocene fossil vertebrates from Burma Quarry, Antigua, Lesser Antilles. *Smithsonian Contribution to Zoology* 463:1–27.

Review form: Reviewer 2

Is the manuscript scientifically sound in its present form?

Yes

Are the interpretations and conclusions justified by the results?

No

Is the language acceptable?

Yes

Do you have any ethical concerns with this paper?

Yes

Have you any concerns about statistical analyses in this paper?

No

Recommendation?

Accept with minor revision (please list in comments)

Comments to the Author(s)

Dear

I have reviewed the manuscript “Unusual lizard fossil from the Miocene of Nebraska and a minimum age for cnemidophorine teiids” (RSOS-200317). The manuscript is original in describing a dentary from the Miocene of the USA, its assignation to the cnemidophorine, would make it the oldest fossil of the clade, and as the authors states it presents a minimum age for them, useful in age calibration studies.

The manuscript is well written (but see below) and accessible to a wide audience. The illustrations are very good and useful. The methodology and methods are appropriate, but I have some consideration on the conclusions that I believe should be addressed.

I have commented directly on the pdf (Appendix A), but my main observations are the following:

- I would recommend the author to improve some passages of the text that are complicated to follow (as the paragraph describing sculpturing in the different taxa). And in other passages it is not clear the phylogenetic hypothesis followed (p6 11, p7 136-26).

- Regarding terminology I would suggest using distal, mesial, labial and lingual for teeth, and posterior, anterior, medial and lateral for the dentary throughout the entire text. As it is the reading of the manuscript is complicated (at least to me).

- In the description please check what is identified as intramandibular septum which seem to be the intramandibular lamella. Please see Denton and O'neill (1995), Lee and Scanlon (2001) and Smith (2009).

- In the description, the author describes implantation as pleurodont, but to me it is subpleurodont (as most other authors consider implantation in teiids). If authors disagrees it should be disused.

- Please see to consider the Barbatteiidae, a European Cretaceous clade considered sister to the extant teiids (Venczel and Codrea 2016; Codrea et al. 2017).

- In the "Diagnosis" section (section should be re-titled). 1) if the tooth presents a "shoulder" than it would be a diagnostic character (I don't know of any Teiidae with bicuspid teeth and a distal shoulder), but I interpret a shoulder as a defined and constant character on a given tooth. From the image this seem to be the mesial cusp with its apex in a more central position than the other preserved teeth, hence the outline. 2) fossil YPM 4607 is lacking the posterior most teeth, which could or not be tricuspid. Presch (1974) states that some Ameiva and Cnemidophorus spp can have only one posterior tricuspid tooth. Therefore in 134-36 bicuspid posterior teeth cannot be consider as a characteristic of the clade.

- In the "Remarks" section. I have noted that in teiids the "lateral eversion" is related to size (development of the adductor musculature)(don't recall if this is mentioned in any bibliography). Did the author compare to different size Holcosus? The specimen in Fig 4 and 5 (UF 51244) is the largest of those presented enforcing this hypothesis. I have doubts on its diagnostic use.

- P8, 18 Cnemidophorus sp MCZ 3381 is from the lower Miocene Florida in (Estes 1963, 1983:90). Both papers cited in manuscript, It would be older? Is so it would be the minimum age of the cnemidophorine clade?

Sincerely Santiago

bibliography suggested (here and pdf)

Brizuela, S., and A. M. Albino. 2009. The dentition of the Neotropical lizard genus *Teius* Merrem 1820 (Squamata Teiidae). *Tropical Zoology* 22:183-193.

Brizuela, S., and R. Kosma. 2017. Comments on the Dentition of the Teiid *Dicrodon* Duméril and Bibron, 1839. *South American Journal of Herpetology* 12:200-204.

Codrea, V. A., M. Venczel, and A. Solomon. 2017. A new family of teioid lizards from the Upper Cretaceous of Romania with notes on the evolutionary history of early teioids. *Zoological Journal of the Linnean Society*.

Lee, M. S. Y., and J. D. Scanlon. 2001. On the lower jaw and intramandibular septum in snakes and anguimorph lizards. *Copeia* 531-535.

Presch, W. 1980. Evolutionary History of the South American Microteiid Lizards (Teiidae: Gymnophthalminae). *Copeia* 1980:36-56.

Smith, K. T. 2009. Eocene Lizards of the Clade Geiseltaliellus from Messel and Geiseltal, Germany, and the Early Radiation of Iguanidae (Reptilia: Squamata). *Bulletin of the Peabody Museum of Natural History* 50:219-306.

Venczel, M., and V. A. Codrea. 2016. A new teiid lizard from the Late Cretaceous of the Hațeg Basin, Romania and its phylogenetic and palaeobiogeographical relationships. *Journal of Systematic Palaeontology* 14:219-237.

Review form: Reviewer 3 (Randall Nydam)

Is the manuscript scientifically sound in its present form?

No

Are the interpretations and conclusions justified by the results?

Yes

Is the language acceptable?

No

Do you have any ethical concerns with this paper?

No

Have you any concerns about statistical analyses in this paper?

No

Recommendation?

Major revision is needed (please make suggestions in comments)

Comments to the Author(s)

What a cool specimen! It is clearly an important contribution to the knowledge of true Teiidae in North America. The scoring in the review is not as bad as it appears, but is the result of insufficient categories and choices to more clearly indicate the areas of success and the areas in need of improvement. I like the work, but I have many well intentioned suggestions for improvement. The anatomical treatment is good, but the language is unnecessarily complicated (see my notes in the manuscript) and can be easily modified to be more accessible and clear. The author is missing some relevant literature such as the recent dental studies of Aaron LeBlanc and the Nydam et al., 2010 JVP article redefining Chamopsiidae. The reason for the request for Major Revision is due to the unfortunate lack of a meaningful exploration of the meaning of this fossil in the fossil record. As the purported oldest cnemidophorine specimen in North America (regardless of method of diagnosis) it provides evidence to address the hypotheses of the timing and mechanism of the arrival of true Teiidae in North America. The addition of such a discussion would dramatically increase the impact of this paper. Please see my comments in the manuscript (Appendix B) and I wish the author the best of luck as he is clearly a very skilled burgeoning paleoherpetologist.

Decision letter (RSOS-200317.R0)

Dear Dr Scarpetta,

The editors assigned to your paper ("Unusual lizard fossil from the Miocene of Nebraska and a minimum age for cnemidophorine teiids") have now received comments from reviewers. We would like you to revise your paper in accordance with the referee and Associate Editor suggestions which can be found below (not including confidential reports to the Editor). Please note this decision does not guarantee eventual acceptance.

Please submit a copy of your revised paper before 21-Jun-2020. Please note that the revision deadline will expire at 00.00am on this date. If we do not hear from you within this time then it will be assumed that the paper has been withdrawn. In exceptional circumstances, extensions may be possible if agreed with the Editorial Office in advance. We do not allow multiple rounds of revision so we urge you to make every effort to fully address all of the comments at this stage. If deemed necessary by the Editors, your manuscript will be sent back to one or more of the original reviewers for assessment. If the original reviewers are not available, we may invite new reviewers.

- Data accessibility

<http://datadryad.org/submit?journalID=RSOS&manu=RSOS-200317>

- Competing interests

- Authors' contributions

- Acknowledgements

- Funding statement

Kind regards,
Lianne Parkhouse
Editorial Coordinator
Royal Society Open Science
openscience@royalsociety.org

on behalf of Dr Emily Lindsey (Associate Editor) and Kevin Padian (Subject Editor)
openscience@royalsociety.org

Associate Editor's comments (Dr Emily Lindsey):

The article has been provided thorough reviews by three experts in the field, and a major revision is suggested. All reviewers agreed this is an interesting discovery that represents an important contribution to the discipline. To make the study acceptable for publication, reviewers have recommended improving the scientific descriptions and interpretations of the study. Key suggestions include:

- Use consistent terminology and reliable measurements in morphological descriptions
- Expand comparisons with other groups (variability in dentary morphology and dentition, ontogenetic evolution, divergence times), including more references to the recent literature
- Provide more information on the fossil context (other taxa known from this formation, etc.)
- Include a better description of the research questions in the Introduction and a greater discussion of the significance of the results in the Discussion

Please see detailed reviews and in-manuscript comments from the three reviewers for additional guidance.

Reviewers' Comments to Author:

Reviewer: 1

Comments to the Author(s)

Dear Author,

Your paper constitutes, in my opinion, an interesting discovery that will for sure help to better understand the history of this poorly known (from a paleontological and osteological point of view) group of lizards.

I am not a specialist of North American squamates so I will not be able to provide much advice regarding most of the taxa discussed in your paper but I however did some work on the osteology of *Pholidoscelis*. As such, I believe that more precision could easily be added in the paper regarding the dentary and teeth morphology of this genus. All my comments regarding this are based on a work recently published regarding the fossil *Pholidoscelis* of Guadeloupe in the framework of which I wrote a short synthesis regarding the teeth morphology and osteological variability of modern *Pholidoscelis*. You seem to have missed this publication (Bochaton et al., 2019) which could maybe help you to bring more precision to the following points:

-I think your description of the ontogenetic evolution of the teeth of *Pholidoscelis* lacks precision and reference to the appropriate literature. This variability was first described by Pregill (Pregill, 1981; Pregill et al., 1988) and I tried to expand the description with specimens of most of the species (Bochaton et al., 2019). The general ontogenetic morphological evolution regarding the posterior teeth is tricuspid->bicuspid->monocuspid->bulbous in Lesser Antillean species. In the Greater Antilles most species "stop" this evolution before reaching the last steps and preserve tricuspid or bicuspid posterior teeth in large/adult specimens. This is however not a strict rule (see my comment on *Ameiva*).

-*Pholidoscelis* dentaries of large specimens do present some dermal ornamentation. Not as on your fossil through but limited to the anterior part of the bone (Bochaton et al., 2019).

-Some large specimens of *Ameiva* present bicuspid posterior/distal teeth. This was observed on a specimen from the MNHN, Paris (Bochaton et al., 2019).

-The number of teeth of your specimen (19) falls into the variability of *Pholidoscelis* (15 to 27) (Bochaton et al., 2019). A comparison between the length and the dental row (absent in the paper?) and the number of teeth would maybe help to discard an attribution to *Pholidoscelis*? These comments led to the conclusion that the fossil described in this study cannot be attributed to *Pholidoscelis*, considering that the pattern of dermal sculpturing present on the fossil was previously stated in the literature as absent in this genus. When I first saw the fossil I have been impressed by the posterior height of its labial margin which is also high in large *Pholidoscelis* (there is ontogenetic variability in this). But your fossil seems very small (10.5mm of dental length following the picture) compared to the member of *Pholidoscelis* in which a similar morphology occurs (20-30mm of dental row length).

Another very important point regarding your interpretation of the fossil: in regard to lesser Antillean "*Ameiva*" (= *Pholidoscelis*), Hedges (2006) indicates a divergence time of 45Ma, so much older than your fossil. This should be discussed as it is in strong contradiction with your interpretation.

Here are also other comments regarding the overall presentation of the study:

Introduction

The introduction seems incomplete and is mostly limited to contextual information. The question of the study is not clearly stated here. What is the aim of publishing this fossil? Why was it studied? The introduction contains sufficient context regarding other finds of fossil teiids in the Americas but this part would be much better with a broader introduction at the start and a proper introduction of the research questions.

Also regarding your references, could you split them regarding the different geographic areas, even in a single sentence? This would make easier to know which author speaks about what.

Geological context

This part also lacks important information in my opinion. That would be great to see at least a rapid presentation of the other taxa found in that formation, squamates, or not. That would also help to better understand why this fossil is important. I discovered in the caption of figure 1 that

this locality was previously studied as there is another reference for it? This should be clearly presented in this part. A reference to Fig. 1 should also be added.

Also, you keep referring to the fact that more material would be needed to identify this fossil but it is unclear whether this material exists/could exist or not. Could you clarify this?

Description

Please add the usual measurements in the description, measuring it on the picture might be a source of error.

I hope you will find these few remarks useful to improve this already very interesting study,

Sincerely,

Corentin Bochaton

References

Bochaton, C., R. Boistel, S. Grouard, I. Ineich, A. Tresset, and S. Bailon. 2019. Evolution, diversity and interactions with past human populations of recently extinct *Pholidoscelis* lizards (Squamata: Teiidae) from the Guadeloupe Islands (French West-Indies). *Historical Biology* 31:140–156.

Pregill, G. K. 1981. Late Pleistocene herpetofaunas from Puerto Rico. University of Kansas Museum of Natural History Miscellaneous Publication 71:1–72.

Pregill, G. K., D. W. Steadman, S. L. Olson, and F. V. Grady. 1988. Late Holocene fossil vertebrates from Burma Quarry, Antigua, Lesser Antilles. *Smithsonian Contribution to Zoology* 463:1–27.

Reviewer: 2

Comments to the Author(s)

I have reviewed the manuscript “Unusual lizard fossil from the Miocene of Nebraska and a minimum age for cnemidophorine teiids” (RSOS-200317). The manuscript is original in describing a dentary from the Miocene of the USA, its assignment to the cnemidophorine, would make it the oldest fossil of the clade, and as the authors states it presents a minimum age for them, useful in age calibration studies.

The manuscript is well written (but see below) and accessible to a wide audience. The illustrations are very good and useful. The methodology and methods are appropriate, but I have some consideration on the conclusions that I believe should be addressed.

I have commented directly on the pdf, but my main observations are the following:

- I would recommend the author to improve some passages of the text that are complicated to follow (as the paragraph describing sculpturing in the different taxa). And in other passages it is not clear the phylogenetic hypothesis followed (p6 l1, p7 l36-26).

- Regarding terminology I would suggest using distal, mesial, labial and lingual for teeth, and posterior, anterior, medial and lateral for the dentary throughout the entire text. As it is the reading of the manuscript is complicated (at least to me).

- In the description please check what is identified as intramandibular septum which seem to be the intramandibular lamella. Please see Denton and O’neill (1995), Lee and Scanlon (2001) and Smith (2009).

- In the description, the author describes implantation as pleurodont, but to me it is subpleurodont (as most other authors consider implantation in teiids). If authors disagrees it should be disused.

- Please see to consider the Barbatteiidae, a European Cretaceous clade considered sister to the extant teiids (Venczel and Codrea 2016; Codrea et al. 2017).

- In the “Diagnosis” section (section should be re-titled). 1) if the tooth presents a “shoulder” than it would be a diagnostic character (I don’t know of any Teiidae with bicuspid teeth and a distal shoulder), but I interpret a shoulder as a defined and constant character on a given tooth. From the image this seem to be the mesial cusp with its apex in a more central position than the other preserved teeth, hence the outline. 2) fossil YPM 4607 is lacking the posterior most teeth, which could or not be tricuspid. Presch (1974) states that some *Ameiva* and *Cnemidophorus* spp can have only one posterior tricuspid tooth. Therefore in l34-36 bicuspid posterior teeth cannot be consider as a characteristic of the clade.

- In the “Remarks” section. I have noted that in teiids the "lateral eversion" is related to size (development of the adductor musculature)(don't recall if this is mentioned in any bibliography). Did the author compare to different size *Holcosus*? The specimen in Fig 4 and 5 (UF 51244) is the largest of those presented enforcing this hypothesis. I have doubts on its diagnostic use.

- P8, l8 *Cnemidophorus* sp MCZ 3381 is from the lower Miocene Florida in (Estes 1963, 1983:90). Both papers cited in manuscript, It would be older? Is so it would be the minimum age of the *cnemidophorine* clade?

Sincerely Santiago

bibliography suggested (here and pdf)

Brizuela, S., and A. M. Albino. 2009. The dentition of the Neotropical lizard genus *Teius* Merrem 1820 (Squamata Teiidae). *Tropical Zoology* 22:183–193.

Brizuela, S., and R. Kosma. 2017. Comments on the Dentition of the Teiid *Dicrodon* Duméril and Bibron, 1839. *South American Journal of Herpetology* 12:200–204.

Codrea, V. A., M. Venczel, and A. Solomon. 2017. A new family of teioid lizards from the Upper Cretaceous of Romania with notes on the evolutionary history of early teioids. *Zoological Journal of the Linnean Society*.

Lee, M. S. Y., and J. D. Scanlon. 2001. On the lower jaw and intramandibular septum in snakes and anguimorph lizards. *Copeia* 531–535.

Presch, W. 1980. Evolutionary History of the South American Microteiid Lizards (Teiidae: *Gymnophthalminae*). *Copeia* 1980:36–56.

Smith, K. T. 2009. Eocene Lizards of the Clade *Geiselaliellus* from Messel and Geiselal, Germany, and the Early Radiation of *Iguanidae* (Reptilia: Squamata). *Bulletin of the Peabody Museum of Natural History* 50:219–306.

Venczel, M., and V. A. Codrea. 2016. A new teiid lizard from the Late Cretaceous of the Hațeg Basin, Romania and its phylogenetic and palaeobiogeographical relationships. *Journal of Systematic Palaeontology* 14:219–237.

Reviewer: 3

Comments to the Author(s)

What a cool specimen! It is clearly an important contribution to the knowledge of true Teiidae in North America. The scoring in the review is not as bad as it appears, but is the result of insufficient categories and choices to more clearly indicate the areas of success and the areas in need of improvement. I like the work, but I have many well intentioned suggestions for improvement. The anatomical treatment is good, but the language is unnecessarily complicated (see my notes in the manuscript) and can be easily modified to be more accessible and clear. The author is missing some relevant literature such as the recent dental studies of Aaron LeBlanc and the Nydam et al., 2010 JVP article redefining *Chamopsiidae*. The reason for the request for Major Revision is due to the unfortunate lack of a meaningful exploration of the meaning of this fossil in the fossil record. As the purported oldest *cnemidophorine* specimen in North America (regardless of method of diagnosis) it provides evidence to address the hypotheses of the timing and mechanism of the arrival of true Teiidae in North America. The addition of such a discussion would dramatically increase the impact of this paper. Please see my comments in the manuscript and I wish the author the best of luck as he is clearly a very skilled burgeoning paleoherpetologist.

Author's Response to Decision Letter for (RSOS-200317.R0)

See Appendix C.

Decision letter (RSOS-200317.R1)

Dear Mr Scarpetta,

It is a pleasure to accept your manuscript entitled "Unusual lizard fossil from the Miocene of Nebraska and a minimum age for cnemidophorine teiids" in its current form for publication in Royal Society Open Science.

Kind regards,
Lianne Parkhouse
Editorial Coordinator
Royal Society Open Science
openscience@royalsociety.org

on behalf of Dr Emily Lindsey (Associate Editor) and Kevin Padian (Subject Editor)
openscience@royalsociety.org

Editor's Comments to Author:

Major revisions were requested on the original submission; the AE is satisfied that the concerns raised by reviewers have been addressed. We think the article is clear, well-reasoned, useful, and appropriate for this journal. Thanks for submitting

Appendix A**ROYAL SOCIETY
OPEN SCIENCE****Unusual lizard fossil from the Miocene of Nebraska and a
minimum age for cnemidophorine teiids**

Journal:	Royal Society Open Science
Manuscript ID	RSOS-200317
Article Type:	Research
Date Submitted by the Author:	27-Feb-2020
Complete List of Authors:	Scarpetta , Simon; University of Texas at Austin John A and Katherine G Jackson School of Geosciences, Jackson School of Geosciences
Subject:	palaeontology < BIOLOGY, evolution < BIOLOGY
Keywords:	Teiidae, fossils, apomorphies, Miocene, divergence time
Subject Category:	Organismal and Evolutionary Biology

**Author-supplied statements**

Relevant information will appear here if provided.

***Ethics***

*Does your article include research that required ethical approval or permits?:*

This article does not present research with ethical considerations

*Statement (if applicable):*

No permissions were required. YPM 4607 is repositied at the Yale Peabody Museum.

***Data***

*It is a condition of publication that data, code and materials supporting your paper are made publicly*
*available. Does your paper present new data?:*

Yes

*Statement (if applicable):*

Electronic supplementary material is available as supplementary file ESM 1 at Royal Society Open
Science.

***Conflict of interest***

I/We declare we have no competing interests

*Statement (if applicable):*

CUST_STATE_CONFLICT :No data available.

***Authors' contributions***

I am the only author on this paper

*Statement (if applicable):*

CUST_AUTHOR_CONTRIBUTIONS_TEXT :No data available.

Unusual lizard fossil from the Miocene of Nebraska and a minimum age for cnemidophorine
teiids

Short title: Miocene teiid fossil from Nebraska

Simon G. Scarpetta

Department of Geological Sciences, Jackson School of Geosciences, The University of Texas at
Austin

Abstract

Teiid lizards are well-represented in the fossil record and are common components of modern ecosystems in North and South America. Many fossils were referred to the cnemidophorine teiid group (whiptails, racerunners, and relatives), particularly from North America. However, the historically problematic taxonomy of cnemidophorines created difficulties interpreting the systematic significance of morphological features in that clade. As a result, few of those fossils were identified with an apomorphy-based diagnosis and there are almost no cnemidophorine teiid fossils that could be used to anchor node calibrations. Here, I describe a cnemidophorine fossil from the Miocene Ogallala Group of Nebraska and diagnose the fossil using apomorphies, and in that process, clarify the systematic utility of several morphological features of teiid lizards. I refer the fossil to the clade *Aspidoscelis* + *Holcosus* + *Pholidoscelis*. The minimum age of the locality of the fossil is 6.3 Ma, which can be used as a minimum age for the crown cnemidophorine clade in divergence time analyses. The fossil has some unusual morphological features compared to other known teiids, but I refrain from naming a new taxon pending discovery of additional material.

Introduction

Teiidae is a clade of diurnal and largely terrestrial New World lizards with a substantial Mesozoic and Cenozoic fossil record, particularly from South America (Albino and Brizuela 2014; Vitt and Pianka 2004). Borioteiioidea, the putative sister clade of Teiidae (Nydham et al. 2007; Simões et al. 2018, but see Daza et al. 2016, Gauthier et al. 2012, Lee 2009, Pyron 2017, Reeder et al. 2015), is known from the Cretaceous of North America and Eurasia. Teiid fossils were recovered in large numbers from Quaternary deposits in mainland North America (north of Mexico) and the Caribbean (e.g., Bell 1993; Bochaton et al. 2015; Parmley and Bahn 2012; Van Devender and Mead 1978), and some fossils were reported from mainland Neogene localities (Bryant 1991; Chovanec 2014; Estes 1963; Estes and Tihen 1964; Holman 1975; Joeckel 1988; Norell 1989; Norell and de Queiroz 1991; Parmley and Peck 2002; Taylor 1941; Tucker et al. 2014; see Fig 1). **There are no known North American teiids from the Paleogene.** North American fossil teiids from the Neogene are mostly isolated and fragmented dentaries and maxillae. The identifications of those fossils were hindered by morphological similarities between and problematic taxonomy of cnemidophorine teiids (Bell 1993; Estes 1963; Norell 1989). The lack of unambiguously identified fossil teiids from the Neogene and **late Paleogene** of North America is remarkable given that extant teiids are widespread and common components of modern North American ecosystems (Vitt and Pianka 2004), divergence time analyses indicate that most crown teiid clades originated during the Oligocene and Miocene (Tucker et al. 2017; Zheng and Wien 2016), and many teiid skeletal elements are distinctly different from those of other squamates (Bell 1993; Estes et al. 1988) and so should be diagnosable through an apomorphy-based approach (Bell et al. 2010).

I describe a nearly complete and unusual fossil dentary of a cnemidophorine teiid lizard from the Miocene Ogallala Group of Cherry County, Nebraska. ~~Although the fossil is a marginal tooth-bearing element,~~ like most other known teiid fossils from the Neogene, it is largely complete and preserves morphologies that allow for a referral to the total clade ~~*Aspidoscelis* + *Holcosus* + *Pholidoscelis*~~. A single fossil referred to cf. *Cnemidophorus* was previously reported

from the Valentine Formation of the Ogallala Group, but the fossil was lost before publication
(Estes and Tihen 1964).

**Materials and Methods**

**Geologic Setting and Collection of Specimen**

The specimen is repositied at the Yale Peabody Museum and was collected by Oscar Harger on
the Yale College scientific expedition of 1873. Unfortunately, that limits the available locality
information to the Ogallala Group at the Niobrara River north of Minnechaduzza Creek in Cherry
County, Nebraska. The Ogallala Group in that region of Cherry County is represented by the
Valentine and Ash Hollow Formations (Skinner and Johnson 1984), constraining the age of the
fossil from 13.5 ± 0.01 Ma based on interpolation of the age of the Hurlbut Ash between dated
horizons (Perkins and Nash 2002) or 13.55 ± 0.09 Ma based on $^{40}\text{Ar}/^{39}\text{Ar}$ dates for the Hurlbut
Ash (Swisher 1992) to 6.6 ± 0.3 Ma based on fission-track of glass from near the top of the Ash
Hollow Formation (Skinner and Johnson 1984; Tedford et al. 2004). The minimum age of the
fossil is 6.3 Ma.

**Institutional Abbreviations**

**CAS** California Academy of Sciences; **MVZ** Museum of Vertebrate Zoology, Herpetology
Collection, University of California, Berkeley; **TNHC** Biodiversity Collections, Herpetology
Collections (Texas Natural History Collections), The University of Texas at Austin; **TxVP** Texas
Vertebrate Paleontology, The University of Texas at Austin (**TMM**); **UF** Florida Museum of
Natural History, Herpetology Division, University of Florida, Gainesville; **YPM** Yale Peabody
Museum.

**Terminology and Taxonomy**

Osteological terminology follows Evans (2008) unless otherwise noted. Taxonomy of
cnemidophorines follows Goicoechea et al. (2013), Harvey et al. (2012), and Reeder et al.
(2002).

**Results**

**Systematic Paleontology**

Teiidae Gray 1827

*Aspidoscelis* Fitzinger 1843; *Holcosus* Cope 1862; *Pholidoscelis* Fitzinger 1843

~~*Aspidoseelis*, *Holcosus* + *Pholidoseelis* sp.~~

Referred specimen: YPM 4607

Figure 2

**Description**

YPM 4607 is a mostly complete right dentary that preserves the tooth row, four teeth, the ramus, and some of the coronoid and angular processes (Fig 2A-B). The distal portion of the dentary is significantly taller than the mesial portion of the dentary, which tapers height rapidly. The Meckelian groove is long and open along the entire element. Mesially, the groove faces ventrally and is restricted by the suprameckelian lip (sensu Bhullar and Smith 2008). The distal portion of the groove has a deep labiolingual dimension, at least in part due to a lateral eversion of the labial surface of the dentary at the coronoid facet (Fig 2C). The mesial extent of the Meckelian groove leaves space for the splenial to extend almost to the symphyseal facet (Fig 2A, D). The suprameckelian lip has a relatively short dorsoventral dimension distally, but markedly increases in height mesially. Distally, the suprameckelian lip possesses inset ventral exposure. Lateral to the suprameckelian lip there is a posterior extension of the intramandibular septum that would separate the medial process of the coronoid and the surangular. The subdental gutter is moderately well-developed.

There are 19 apparent tooth positions, but it is difficult to distinguish the mesialmost tooth positions, so that count should be viewed as uncertain. Three distal teeth and one mesial tooth are preserved, and the three distal teeth are spaced somewhat far from each other. Dentition is pleurodont and heterodont in terms of both tooth size and cusp morphology. The distal-most tooth has a mesiodistally expanded base, and the two teeth mesial to that tooth taper less distinctly from the base to the crown. The crowns of the three distal teeth are asymmetrically and longitudinally bicuspid, and the mesial crown is smaller than the main crown. The main (distal) crown of the third tooth slopes ventrodistally instead of ventrally to create a slight distal shoulder. The mesial crowns of all three bicuspid teeth are apically worn. The mesial tooth is unicuspid and is substantially smaller than the distal teeth. A replacement pit is present on the second tooth, but pits are present within dental tissues at other tooth positions that lack teeth. The replacement pits are deep and almost circular. The teeth have substantial but not excessive deposits of basal cementum.

The labial face of the dentary is moderately convex and has rugose sculpturing across much of its surface, but rugosities are absent immediately ventral to the three distal teeth and on the mesial-most portion of the element (Fig 2B). The coronoid facet is subtriangular and extends anteriorly to the fourth tooth, and ventrally to just above the dorsoventral level of a row of nutrient foramina. The facet is deep, moderately textured, and has a marked ventral boundary. There are nine distinct nutrient foramina that extend distally to the second tooth, although most of the foramina are concentrated mesially. The distal portion of the dentary is less complete than the rest of the fossil, and so the posterior extent of the coronoid and angular processes is uncertain, as is the presence of a separate surangular process. However, the coronoid process does not appear to have a large dorsal extent.

Diagnosis

The presence of pleurodont teeth that are superficially attached to the lingual surface of the jaw diagnoses YPM 4607 as a member of Lepidosauria (Estes et al. 1988; Gauthier et al. 1988). Rhynchocephalians generally have acrodont teeth, although *Gephyrosaurus* has more apically positioned pleurodont teeth (Evans 1980).

Asymmetrically bicuspid distal teeth and an open Meckelian groove are characteristic of Teiidae, but are also present in Lacertidae, which is closely related to Teiidae.

Gymnophthalmidae + Alopoglossidae, the sister clade of Teiidae, can also have bicuspid teeth, but the Meckelian groove is generally closed and fused (Bell et al. 2003). Amphisbaenia, the sister taxon of Lacertidae, possesses several unique derived features, including unicuspid teeth, a short dentary, a low tooth count (<10 teeth), and acrodont, subacrodont, or **subpleurodont** dentition (Gans et al. 2008; Longrich et al. 2015). YPM 4607 has substantial deposits of cementum at its tooth bases, large subcircular tooth replacement pits, and an elongate Meckelian groove providing space for a hypertrophied splenial, all of which are apomorphies that diagnose Teiidae to the exclusion of other squamates (Estes et al. 1988, Nydam et al. 2007).

Borioteioids can have all three dentary apomorphies commonly used to diagnose teiid fossils (Nydam et al. 2007, see above) that are present in YPM 4607. The fossil is excluded from Polyglyphanodontia because it lacks posterior teeth with transversely-oriented cusps, from *Chamops* because the Meckelian groove is anteriorly restricted, there are no distinctly tricuspid teeth, and the distal teeth are markedly less bulbous (Estes 1983), from *Prototeius* because the posterior teeth are bicuspid and are not massive or blunt-cusped and the Meckelian groove is anteriorly restricted (Denton and O'Neill 1995), and from members of Macrocephalosauridae, which have flared, multicuspid teeth, sometimes with obliquely-oriented cusps (Sulimski 1975). YPM 4607 is part of total clade Teiidae *sensu stricto* (excluding Borioteioidea).

The mesial restriction of the Meckelian groove excludes the fossil from Tupinambinae, whose members have a completely unrestricted and medially facing Meckelian groove (Denton and O'Neill 1995; Albino et al. 2013), but is a feature shared with Teiinae (Fig 3). Tupinambines are also distinguished from the fossil by possessing a **massive** incision in between the coronoid and angular processes of the dentary. The teiines *Dicrodon* and *Teius* have more basal cementum relative to cnemidophorines and YPM 4607 (Fig 3G; Albino et al. 2013). The fossil also lacks the transverse tooth cusps of *Dicrodon* (Fig 3G, 4G, 5G) and *Teius* (Presch 1974).

Among examined teiids (a complete list of specimens is in ESM 1), only *Aspidoscelis* and *Holcosus* have sculpturing on the labial surface of the dentary (Fig 4A-C). Labial sculpturing was reported to occur in adult individuals of *Ameiva* and *Cnemidophorus* without reference to any particular species (Norell 1989) and before those genera were split up. Analyses by Tucker et al. (2016) and Tucker et al. (2017) of an anchored phylogenomic dataset inferred *Aspidoscelis* and *Holcosus* as sister taxa. Given that hypothesis, sculpturing on the dentary appears to be an apomorphy of the *Aspidoscelis* + *Holcosus* clade. *Aspidoscelis* + *Pholidoscelis* were estimated as sister taxa and *Holcosus* was sister to that clade in the analysis of Zheng and Wiens (2016), but those relationships were supported by negligible bootstrap values. That relationship would imply a single origin for sculpturing for the entire *Aspidoscelis* + *Pholidoscelis* + *Holcosus* clade or independent origins in *Aspidoscelis* and *Holcosus* with a loss in *Pholidoscelis*. Alternatively, *Aspidoscelis*, *Holcosus*, and *Pholidoscelis* were paraphyletic with respect to each other within Teiinae in the analyses of Goicoechea et al. (2016). **I accept that *Aspidoscelis*, *Holcosus*, and *Pholidoscelis* form a clade to the exclusion of other extant teiid lizards, following Tucker et al. (2016), Tucker et al. (2017), and Zheng and Wiens (2016).**

Most specimens of *Aspidoscelis* lack sculpturing. In both *Aspidoscelis* and *Holcosus*, sculpturing occurs often but not universally on larger and more robust specimens that were probably skeletally mature adults, as was observed by Norell (1989). *Holcosus festiva* MVZ 79608 is a larger specimen that lacks any sculpturing. Sculpturing is present on all examined specimens of *Aspidoscelis deppii*, some specimens of *Aspidoscelis tigris* (e.g., TxVP M-8629, M-8631, M-15034), *Aspidoscelis gularis* TxVP M-15028, and on *Holcosus quadrilineatus* and *Holcosus undulatus* UF 51244. In *Holcosus*, labial sculpturing ranges from longitudinal and

wispy texturing (*Holcosus undulatus*; Fig 4C) to more pronounced and vermiculate sculpturing (*Holcosus quadrilineatus* UF 37170). In *Aspidoscelis*, labial sculpturing can be rugose (*Aspidoscelis tigris*; Fig 4B) or less complexly textured (*Aspidoscelis deppii*; Fig. 4A). The sculpturing of YPM 4607 is most similar to examined specimens of *Aspidoscelis tigris*. I did not find sculpturing on specimens of *Ameiva*, *Cnemidophorus*, *Kentropyx*, *Medopheos*, or *Pholidoscelis*. I sampled all cnemidophorine genera, but sampling of more species, particularly of *Pholidoscelis*, is desirable to further establish the distribution of sculpturing within cnemidophorines. I observed no sculpturing in tupinambines, *Dicrodon*, or *Teius*, and I did not examine the teiines *Aurivela*, *Contomastix*, *Glaucomastix*, or *Ameivula*.

While tooth cusp morphology is known to vary ontogenetically and interspecifically in teiids (Estes and Williams 1984), all examined specimens of *Holcosus*, *Ameiva*, *Cnemidophorus*, *Kentropyx*, and *Medopheos* have exclusively tricuspid distal dentary teeth, as do *Aurivela*, *Contomastix*, and *Ameivula* (see Estes 1963, Harvey et al. 2012, Presch 1974). In *Holcosus*, *Ameiva*, *Cnemidophorus*, *Kentropyx*, and *Medopheos*, the transition from unicuspid teeth to tricuspid teeth occurs on the mesial half of the tooth row, often abruptly. Many *Aspidoscelis* have bicuspid teeth throughout the tooth row, but some species, such as *Aspidoscelis uniparens* and *Aspidoscelis sonora*, have mostly tricuspid teeth. Some small specimens of *Pholidoscelis* have mostly tricuspid distal teeth (e.g., *Pholidoscelis chrysolaeus* UF 99352), but in larger specimens distal teeth are all bicuspid (Fig 3D, 4D) or are mostly bicuspid except for the teeth at the last two or three tooth positions, which are tricuspid (e.g., *Pholidoscelis chrysolaeus* UF 99646). Having bicuspid distal teeth is either a separately derived state in *Aspidoscelis* and *Pholidoscelis* with respect to other cnemidophorines, or is derived in the clade *Aspidoscelis* + *Holcosus* + *Pholidoscelis*, with an apparent reversal in *Holcosus*. The third tooth of YPM 4607 is sloped ventrodistally to create a slight dorsal shoulder on the distal face of the tooth. That morphology also occurs in some specimens of *Aspidoscelis* that have bicuspid distal teeth (e.g., *Aspidoscelis deppii*, Fig 3A, 4A). I interpret the third tooth of YPM 4607 as representing a shoulder instead of a third cusp, but recognize that there is some ambiguity.

The morphology of the Meckelian groove, labial sculpturing, and the presence of bicuspid distal teeth place YPM 4607 in total clade *Aspidoscelis* + *Holcosus* + *Pholidoscelis*. Based on the available material, labial sculpturing is unique to *Aspidoscelis* and *Holcosus*; however, given that I examined only four of the twenty species of *Pholidoscelis*, I refrain from excluding that clade for the time being and instead refer the fossil to total clade *Aspidoscelis* + *Holcosus* + *Pholidoscelis*.

Remarks

YPM 4607 is unusual among teiids in having a strongly and abruptly tapered anterior portion of the dentary. No examined teiids have a comparable morphology, although I observed moderately stepped tapering of the dentary in *Cnemidophorus lemniscatus* (Fig 3F, 4F), *Aspidoscelis deppii* (Fig 3A, 4A), *Aspidoscelis tigris* (e.g., TxVP M-15034, M-15035), and *Holcosus undulatus* (Fig 3C, 4C). Lateral eversion of the labial surface of the dentary is most pronounced in *Holcosus* among extant teiids (Fig 5C). Correspondingly, the ventral border of the coronoid facet is most distinct in *Holcosus*. The facet has a less defined ventral border in *Aspidoscelis*, *Pholidoscelis*, and other cnemidophorines, and eversion of the dentary is less exaggerated in those taxa as well. Eversion of the dentary is comparable between YPM 4607 and *Holcosus*.

Two of the four examined species of *Holcosus* had dentary sculpturing. The unusually complex and incompletely resolved phylogeny of *Aspidoscelis* (Barley et al. 2019; Reeder et al. 2002) and intraspecific and intra-clade variability in dentary sculpturing makes the evolution of that morphology difficult to interpret within *Aspidoscelis*. Some specimens of *Aspidoscelis tigris* have highly reduced sculpturing, including larger specimens (e.g., TxVP M-8630). *Aspidoscelis hyperthyrus*, which is closely related to *Aspidoscelis deppii*, lacks sculpturing altogether.

The presence of a moderately developed distal extension of the intramandibular septum (IMS) was reported to diagnose polyglyphanodontids to the exclusion of extant teiids (Denton and O'Neill 1995). However, I observe a well-developed distal IMS in several extant teiids, including *Aspidoscelis* and *Holcosus* (Fig 3A-C).

Discussion

To my knowledge, YPM 4607 is the oldest published record of a crown cnemidophorine from North America identified explicitly using apomorphies. The minimum age of total clade *Aspidoscelis* + *Holcosus* + *Pholidoscelis* and the minimum age of the crown cnemidophorine clade is 6.6 ± 0.3 Ma, based on a fission track age estimate of glass from the Ash Hollow Formation (Skinner and Johnson 1984). Thus, 6.3 Ma can be used as a minimum age for the divergence between the *Aspidoscelis* + *Holcosus* + *Pholidoscelis* clade and the *Ameiva* + *Cnemidophorus* + *Kentropyx* + *Medopheos* clade (i.e., the age of crown cnemidophorines). Few fossils were previously considered reliable for bracketing minimum clade ages in divergence time analyses of Teiidae, and only one cnemidophorine of indeterminate phylogenetic affinity was used recently (Albino et al. 2013; Tucker et al. 2017). YPM 4607 is significant for future studies seeking to temporally calibrate the evolutionary history of teiids and cnemidophorines in particular.

The resurrection of the genera *Aspidoscelis*, *Holcosus*, and *Pholidoscelis* for clades previously accommodated in *Ameiva* and *Cnemidophorus* (Goicoechea et al. 2016; Harvey et al. 2012; Reeder et al. 2002) was helpful here for clarifying the systematic significance of two morphologies in cnemidophorines. Labial sculpturing of the dentary was reported in adult specimens of *Ameiva* and *Cnemidophorus* by Norell (1989). Given the sample here, sculpturing is restricted to two clades formerly in *Ameiva* and *Cnemidophorus*, *Holcosus* and *Aspidoscelis*, that are hypothesized to be each other's closest relatives (Tucker et al. 2016; 2017). Similarly, bicuspid distal tooth crowns were previously reported in certain groups of *Ameiva* and *Cnemidophorus* (Presch 1974; Estes 1963). The West Indian *Ameiva* (i.e. *Pholidoscelis*) were recognized as the only *Ameiva* to have bicuspid teeth by Estes (1963), but no pattern was perceived in *Cnemidophorus*. *Pholidoscelis* and *Aspidoscelis*, which are closely related, are the only genera previously assigned to *Ameiva* and *Cnemidophorus* that have bicuspid distal teeth. Tooth cusp morphology has been an enigmatic feature to interpret in cnemidophorine teiids, but now appears to be elucidated systematically. The presence of dentary sculpturing and bicuspid teeth are apomorphies that can be used to identify cnemidophorine fossils in the future, and the presence or absence of those features could potentially be used as characters in phylogenetic analyses of morphological data.

Aspidoscelis sexlineatus is the only teiid that occurs in Nebraska or immediately adjacent to Nebraska in the modern biota (Stebbins 2003), indicating that a different lineage of cnemidophorine was found in the area at least as recently as the middle-late Miocene. *Aspidoscelis tigris*, the *Aspidoscelis tessellatus* complex, the *Aspidoscelis neotessellatus* complex,

and *Aspidoscelis velox* are currently found in southern and western Colorado (Stebbins 2003).
*Pholidoscelis* is currently restricted to islands in the Caribbean, and *Holcosus* is found in Central
and South America and in Mexico as far north as central Tamaulipas (Goicoechea et al. 2016;
Lavín-Murcio and Lazcano 2010). More fossils from across North America are needed to
explore the historical biogeography of cnemidophorine lizards and to determine whether modern
tropical or island clades like *Holcosus* and *Pholidoscelis* were once found farther north on the
mainland or on the mainland at all, respectively.

The present survey of extant teiids indicates that the overall morphotype of YPM 4607 is
unique, particularly the mesial tapering of the dentary. No teiid taxon found in the extant biota
has a directly comparable set of features. However, the fossil cannot be definitively excluded
from total or crown *Aspidoscelis*, *Pholidoscelis*, or *Holcosus*, so for now I refrain from
establishing a new taxon should new material reveal that YPM 4607 is referable to one of those
clades. Additional fossils of the lineage to which YPM 4607 belonged are needed to refine its
systematic position, investigate whether other skeletal elements were similarly distinctive, and
determine whether a new taxon is warranted.

**Ethics**

No permissions were required. YPM 4607 is deposited at the Yale Peabody Museum.

**Data accessibility**

Electronic supplementary material is available as supplementary file ESM 1 at *Royal Society*
*Open Science*.

32 **Competing interests**

33
34 The author declares no competing interests.

36 **Funding**

[revised manuscript text omitted]

Chovanec K. 2014 Non-anguimorph lizards of the late Oligocene and early Miocene of Florida
and implications for the reorganization of the North American herpetofauna. M.Sc. Thesis,
Department of Geosciences, East Tennessee State University. 123 pp. Available from
<https://dc.etsu.edu/cgi/viewcontent.cgi?article=3732&context=etd>. Accessed 23 June 2016.

Cope ED. 1862 Synopsis of the species of *Holcosus* and *Ameiva*, with diagnoses of new West
Indian and South American Colubridæ. *Proceedings of the Academy of Natural Sciences of*
*Philadelphia* **14**, 60–82.

Daza JD, Stanley EL, Wagner P, Bauer AM, Grimaldi, DA. 2016 Mid-Cretaceous amber fossils
illuminate the past diversity of tropical lizards. *Science Advances* **2**, e1501080.
<https://doi.org/10.1126/sciadv.1501080>

Denton Jr RK, O'Neill RC. 1995 *Prototeius stageri*, Gen. et sp. Nov., a new Teiid lizard from
the Upper Cretaceous Marshalltown Formation of New Jersey, with a preliminary phylogenetic
revision of the Teiidae. *Journal of Vertebrate Paleontology* **15**, 235–253.

Estes R. 1963 Early Miocene salamanders and lizards from Florida. *Quarterly Journal of Florida*
*Academy of Sciences* **25**, 234–256.

Estes R. 1983 *Encyclopedia of Paleoherpetology, Sauria terrestria, Amphisbaenia*. Gustav Fisher
Verlag, Stuttgart, Germany, xxii+249 pp.

Estes R, Tihen JA. 1964 Lower vertebrates from the Valentine Formation of Nebraska. *The*
*American Midland Naturalist* **72**, 453–472.

Estes R, Williams EE. 1984 Ontogenetic variation in the molariform teeth of lizards. *Journal of*
*Vertebrate Paleontology* **4**, 96–107.

Estes R, De Queiroz K, Gauthier J. 1988 Phylogenetic relationships within Squamata. In
Phylogenetic Relationships of the Lizard Families: Essays Commemorating Charles L. Camp
(pp. 119–281). Stanford, California: Stanford University Press.

Evans SE. 1980 The skull of a new eosuchian reptile from the Lower Jurassic of South Wales.
Zoological Journal of the Linnean Society **70**, 203–264. [https://doi.org/10.1111/j.1096-](https://doi.org/10.1111/j.1096-3642.1980.tb00852.x)
[3642.1980.tb00852.x](https://doi.org/10.1111/j.1096-3642.1980.tb00852.x)

Fitzinger L. 1843 Systema Reptilium, fasciculus primus, Amblyglossae. Braumüller et Seidel,
Wien: 106 pp.

Gans C, Montero R. 2008 An atlas of amphisbaenian skull anatomy. In *Biology of the Reptilia,*
*Volume 21* (eds C Gans, AS Gaunt, K Adler), pp. 621–738. Ithaca, NY: Society for the Study of
Amphibians and Reptiles.

Gauthier J, Estes R, de Queiroz K. 1988 A phylogenetic analysis of Lepidosauromorpha. In
*Phylogenetic Relationships of the Lizard Families: Essays Commemorating Charles L. Camp*
(eds R Estes, GK Pregill), pp. 15–98. Stanford, CA: Stanford University Press.

Gauthier JA, Kearney M, Maisano JA, Rieppel O, Behlke ADB. 2012 Assembling the squamate
tree of life: Perspectives from the phenotype and the fossil record. *Bulletin of the Peabody*
*Museum of Natural History* **53**, 3–308. <https://doi.org/10.3374/014.053.0101>

Goicoechea N, Frost DR, Riva I, Pellegrino KCM, Sites JJ, Rodrigues MT, Padial JM. 2016
Molecular systematics of teioid lizards (Teioidea/Gymnophthalmoidea: Squamata) based on the
analysis of 48 loci. *Cladistics* **32**, 1–48. <https://doi.org/https://doi.org/10.1111/cla.12150>

Gray JE. 1827 A synopsis of the genera of Saurian reptiles, in which some new genera are
indicated, and the others reviewed by actual examination. *The Philosophical Magazine* **2**, 54–58.
<https://doi.org/10.1080/14786442708675620>

Harvey MB, Ugueto GN, Gutberlet RL. 2012 Review of teiid morphology with a revised
taxonomy and phylogeny of the Teiidae (Lepidosauria: Squamata). *Zootaxa* **3459**, 1–156.

Holman JA. 1975 Herpetofauna of the WaKeeney Local Fauna (Lower Pliocene: Clarendonian)
of Trego County, Kansas. *University of Michigan Papers in Paleontology* **3**, 49–66.

Joeckel RM. 1988 A new late Miocene herpetofauna from Franklin County, Nebraska. *Copeia*
**1988**, 787–789.

Lavín-Murcio PA, Lazcano DA. 2010 Geographic distribution and conservation of the
herpetofauna of Northern Mexico. In *Conservation of Mesoamerican reptiles and amphibians*
(eds LD Wilson, JH Townsend, JD Johnson), pp. 274–301. Eagle Mountain, UT: Eagle Mountain
Publishing, LC.

Lee MSY. 2009 Hidden support from unpromising data sets strongly unites snakes with
anguimorph “lizards.” *Journal of Evolutionary Biology* **22**, 1308–1316.
<https://doi.org/10.1111/j.1420-9101.2009.01751.x>
Longrich NR, Vinther J, Pyron RA, Pisani D, Gauthier, JA. 2015 Biogeography of worm lizards
(Amphisbaenia) driven by end-Cretaceous mass extinction. *Proceedings of the Royal Society B*
**282**, 1–10. <https://doi.org/10.1098/rspb.2014.3034>
Norell MA. 1989 Late Cenozoic lizards of the Anza Borrego Desert, California. *Contributions in*
*Science, Natural History Museum of Los Angeles County* **414**, 1–31.
Norell, MA, Queiroz K De. 1991 The earliest iguanine lizard (Reptilia: Squamata) and its
bearing on iguanine phylogeny. *American Museum Novitates* **2997**, 1–16.
Nydam ARL, Eaton JG, Sankey J. 2007 New taxa of transversely-toothed lizards (Squamata:
Scincomorpha) and new information on the evolutionary history of “teiids.” *Journal of*
*Paleontology* **81**, 538–549.
Parmley D, Peck D. 2002 Amphibians and reptiles of the late Hemphillian White Cone local
fauna, Navajo County, Arizona. *Journal of Vertebrate Paleontology* **22**, 175–178.
[https://doi.org/10.1671/0272-4634\(2002\)022\[0175:AAROTL\]2.0.CO;2](https://doi.org/10.1671/0272-4634(2002)022[0175:AAROTL]2.0.CO;2)
Parmley D, Bahn, JR. 2012 Late Pleistocene lizards from Fowlkes Cave, Culberson County,
Texas. *The Southwestern Naturalist* **57**, 435–441. <https://doi.org/10.1894/0038-4909-57.4.435>
Perkins ME, Nash BP. 2002 Explosive silicic volcanism of the Yellowstone hotspot: The ash fall
tuff record. *Bulletin of the Geological Society of America* **114**, 367–381.
[https://doi.org/10.1130/0016-7606\(2002\)114<0367:ESVOTY>2.0.CO;2](https://doi.org/10.1130/0016-7606(2002)114<0367:ESVOTY>2.0.CO;2)
Presch W. 1974 A survey of the dentition of the macroteiid lizards (Teiidae: Lacertilia).
*Herpetologica* **30**, 344–349.
Pyron RA. 2017 Novel approaches for phylogenetic inference from morphological data and
total-evidence dating in squamate reptiles (lizards, snakes, and amphisbaenians). *Systematic*
*Biology* **66**, 38–56. <https://doi.org/10.1093/sysbio/syw068>
Reeder TW, Cole CJ., Dessauer HC. 2002 Phylogenetic relationships of whiptail lizards of the
genus *Cnemidophorus* (Squamata: Teiidae): A test of monophyly, reevaluation of karyotypic
evolution, and review of hybrid origins. *American Museum Novitates* **3365**, 1–61.
[https://doi.org/10.1206/0003-0082\(2002\)365<0001:prowlo>2.0.co;2](https://doi.org/10.1206/0003-0082(2002)365<0001:prowlo>2.0.co;2)
Reeder TW, Townsend TM, Mulcahy DG, Noonan BP, Wood Jr PL, Sites Jr JW, Wiens JJ. 2015
Integrated analyses resolve conflicts over squamate reptile phylogeny and reveal unexpected
placements for fossil taxa. *PLoS ONE* **10**, 1–22. <http://dx.doi.org/10.1371/journal.pone.0118199>

RStudio Team. 2015 RStudio: Integrated Development for R. RStudio, Inc., Boston, MA.
<http://www.rstudio.com/>

Simões TR, Caldwell MW, Talanda M, Bernardi M, Palci A, Vernygora O, Bernardini F,
Mancini L, Nydam RL. 2018. The origin of squamates revealed by a Middle Triassic lizard from
the Italian Alps. *Nature* **557**, 706–709. <https://doi.org/10.1038/s41586-018-0093-3>

Skinner M, Johnson F. 1984 Tertiary stratigraphy and the Frick Collection of fossil vertebrates
from north-central Nebraska. *Bulletin of the American Museum of Natural History* **178**, i + 217-
368.

Stebbins RC. 2003 A field guide to western reptiles and amphibians. 3rd ed. New York, NY:
Houghton Mifflin Company.

Sulimski A. 1975 Macrocephalosauridae and Polyglyphanodontidae (Sauria) from the Late
Cretaceous of Mongolia. *Palaeontologia Polonica* **33**, i + 26-102 + plates VIII-XXVII.

Swisher CC. 1992 ⁴⁰Ar/³⁹Ar dating and its application to the calibration of the North American
land mammal ages. PhD Dissertation, University of California, Berkeley. 239 pp.

Taylor EH. 1941 Extinct lizards from upper Pliocene deposits of Kansas. *State Geological*
*Survey of Kansas Bulletin* **38**, 165–176.

Tedford RH, Alrbright LBA, Barnosky AD, Ferrusquia-Villafranca I, Hunt RMJ, Storer JE,
Swisher CC, Voorhies MR, Webb SD, Whistler DP. 2004 Mammalian biochronology of the
Arikareean through Hemphillian interval (late Oligocene through early Pliocene epochs). In *Late*
*Cretaceous and Cenozoic Mammals of North America: Biostratigraphy and Geochronology* (ed
MO Woodburne), pp. 169-231. New York, NY: Columbia University Press.

Tennekes M. 2018 “tmap: Thematic Maps in R.” *Journal of Statistical Software* **84**, 1–39.
<http://dx.doi.org/10.18637/jss.v084.i06>

[revised manuscript text omitted]

Journal:	Royal Society Open Science
Manuscript ID	RSOS-200317
Article Type:	Research
Date Submitted by the Author:	27-Feb-2020
Complete List of Authors:	Scarpetta , Simon; University of Texas at Austin John A and Katherine G Jackson School of Geosciences, Jackson School of Geosciences
Subject:	palaeontology < BIOLOGY, evolution < BIOLOGY
Keywords:	Teiidae, fossils, apomorphies, Miocene, divergence time
Subject Category:	Organismal and Evolutionary Biology

Author-supplied statements

Relevant information will appear here if provided.

Ethics

Does your article include research that required ethical approval or permits?:

This article does not present research with ethical considerations

Statement (if applicable):

No permissions were required. YPM 4607 is repositied at the Yale Peabody Museum.

Data

It is a condition of publication that data, code and materials supporting your paper are made publicly available. Does your paper present new data?:

Yes

Statement (if applicable):

Electronic supplementary material is available as supplementary file ESM 1 at Royal Society Open Science.

Conflict of interest

I/We declare we have no competing interests

Statement (if applicable):

CUST_STATE_CONFLICT :No data available.

Authors' contributions

I am the only author on this paper

Statement (if applicable):

CUST_AUTHOR_CONTRIBUTIONS_TEXT :No data available.

Unusual lizard fossil from the Miocene of Nebraska and a minimum age for cnemidophorine
teiids

Short title: Miocene teiid fossil from Nebraska

Simon G. Scarpetta

Department of Geological Sciences, Jackson School of Geosciences, The University of Texas at
Austin

Abstract

Teiid lizards are well-represented in the fossil record and are common components of modern ecosystems in North and South America. Many fossils were referred to the cnemidophorine teiid group (whiptails, racerunners, and relatives), particularly from North America. However, the historically problematic taxonomy of cnemidophorines created difficulties interpreting the systematic significance of morphological features in that clade. As a result, few of those fossils were identified with an apomorphy-based diagnosis and there are almost no cnemidophorine teiid fossils that could be used to anchor node calibrations. Here, I describe a cnemidophorine fossil from the Miocene Ogallala Group of Nebraska and diagnose the fossil using apomorphies, and in that process, clarify the systematic utility of several morphological features of teiid lizards. I refer the fossil to the clade *Aspidoscelis* + *Holcosus* + *Pholidoscelis*. The minimum age of the locality of the fossil is 6.3 Ma, which can be used as a minimum age for the crown cnemidophorine clade in divergence time analyses. The fossil has some unusual morphological features compared to other known teiids, but I refrain from naming a new taxon pending discovery of additional material.

Introduction

Teiidae is a clade of diurnal and largely terrestrial New World lizards with a substantial Mesozoic and Cenozoic fossil record, particularly from South America (Albino and Brizuela 2014; Vitt and Pianka 2004). Borioteiioidea, the putative sister clade of Teiidae (Nydham et al. 2007; Simões et al. 2018, but see Daza et al. 2016, Gauthier et al. 2012, Lee 2009, Pyron 2017, Reeder et al. 2015), is known from the Cretaceous of North America and Eurasia. Teiid fossils were recovered in large numbers from Quaternary deposits in mainland North America (north of Mexico) and the Caribbean (e.g., Bell 1993; Bochaton et al. 2015; Parmley and Bahn 2012; Van Devender and Mead 1978), and some fossils were reported from mainland Neogene localities (Bryant 1991; Chovanec 2014; Estes 1963; Estes and Tihen 1964; Holman 1975; Joeckel 1988; Norell 1989; Norell and de Queiroz 1991; Parmley and Peck 2002; Taylor 1941; Tucker et al. 2014; see Fig 1). There are no known North American teiids from the Paleogene. North American fossil teiids from the Neogene are mostly isolated and fragmented dentaries and maxillae. The identifications of those fossils were hindered by morphological similarities between and problematic taxonomy of cnemidophorine teiids (Bell 1993; Estes 1963; Norell 1989). The lack of unambiguously identified fossil teiids from the Neogene and late Paleogene of North America is remarkable given that extant teiids are widespread and common components of modern North American ecosystems (Vitt and Pianka 2004), divergence time analyses indicate that most crown teiid clades originated during the Oligocene and Miocene (Tucker et al. 2017; Zheng and Wiens 2016), and many teiid skeletal elements are distinctly different from those of other squamates (Bell 1993; Estes et al. 1988) and so should be diagnosable through an apomorphy-based approach (Bell et al. 2010).

I describe a nearly complete and unusual fossil dentary of a cnemidophorine teiid lizard from the Miocene Ogallala Group of Cherry County, Nebraska. Although the fossil is a marginal tooth-bearing element, like most other known teiid fossils from the Neogene, it is largely complete and preserves morphologies that allow for a referral to the total clade *Aspidoscelis* + *Holcosus* + *Pholidoscelis*. A single fossil referred to cf. *Cnemidophorus* was previously reported

from the Valentine Formation of the Ogallala Group, but the fossil was lost before publication
(Estes and Tihen 1964).

**Materials and Methods**

**Geologic Setting and Collection of Specimen**

The specimen is repositied at the Yale Peabody Museum and was collected by Oscar Harger on
the Yale College scientific expedition of 1873. Unfortunately, that limits the available locality
information to the Ogallala Group at the Niobrara River north of Minnechaduza Creek in Cherry
County, Nebraska. The Ogallala Group in that region of Cherry County is represented by the
Valentine and Ash Hollow Formations (Skinner and Johnson 1984), constraining the age of the
fossil from 13.5 ± 0.01 Ma based on interpolation of the age of the Hurlbut Ash between dated
horizons (Perkins and Nash 2002) or 13.55 ± 0.09 Ma based on $^{40}\text{Ar}/^{39}\text{Ar}$ dates for the Hurlbut
Ash (Swisher 1992) to 6.6 ± 0.3 Ma based on fission-track of glass from near the top of the Ash
Hollow Formation (Skinner and Johnson 1984; Tedford et al. 2004). The minimum age of the
fossil is 6.3 Ma.

**Institutional Abbreviations**

**CAS** California Academy of Sciences; **MVZ** Museum of Vertebrate Zoology, Herpetology
Collection, University of California, Berkeley; **TNHC** Biodiversity Collections, Herpetology
Collections (Texas Natural History Collections), The University of Texas at Austin; **TxVP** Texas
Vertebrate Paleontology, The University of Texas at Austin (TMM); **UF** Florida Museum of
Natural History, Herpetology Division, University of Florida, Gainesville; **YPM** Yale Peabody
Museum.

**Terminology and Taxonomy**

Osteological terminology follows Evans (2008) unless otherwise noted. Taxonomy of
cnemidophorines follows Goicoechea et al. (2016), Harvey et al. (2012), and Reeder et al.
(2002).

**Results**

**Systematic Paleontology**

Teiidae Gray 1827

*Aspidoscelis* Fitzinger 1843; *Holcosus* Cope 1862; *Pholidoscelis* Fitzinger 1843

*Aspidoscelis* + *Holcosus* + *Pholidoscelis* sp.

Referred specimen: YPM 4607

Figure 2

**Description**

YPM 4607 is a mostly complete right dentary that preserves the tooth row, four teeth, the ramus,
and some of the coronoid and angular processes (Fig 2A-B). The distal portion of the dentary is
significantly taller than the mesial portion of the dentary, which tapers in height rapidly. The
Meckelian groove is long and open along the entire element. Mesially, the groove faces ventrally
and is restricted by the suprameckelian lip (sensu Bhullar and Smith 2008). The distal portion of
the groove has a deep labiolingual dimension, at least in part due to a lateral eversion of the
labial surface of the dentary at the coronoid facet (Fig 2C). The mesial extent of the Meckelian
groove leaves space for the splenial to extend almost to the symphyseal facet (Fig 2A, D). The
suprameckelian lip has a relatively short dorsoventral dimension distally, but markedly increases
in height mesially. Distally, the suprameckelian lip possesses an inset ventral exposure. Lateral to
the suprameckelian lip there is a posterior extension of the intramandibular septum that would
separate the medial process of the coronoid and the surangular. The subdental gutter is
moderately well-developed.

There are 19 apparent tooth positions, but it is difficult to distinguish the mesialmost
tooth positions, so that count should be viewed as uncertain. Three distal teeth and one mesial
tooth are preserved, and the three distal teeth are spaced somewhat far from each other. Dentition
is pleurodont and heterodont in terms of both tooth size and cusp morphology. The distal-most
tooth has a mesiodistally expanded base, and the two teeth mesial to that tooth taper less
distinctly from the base to the crown. The crowns of the three distal teeth are asymmetrically and
longitudinally bicuspid, and the mesial crown is smaller than the main crown. The main (distal)
crown of the third tooth slopes ventrodistally instead of ventrally to create a slight distal
shoulder. The mesial crowns of all three bicuspid teeth are apically worn. The mesial tooth is
unicuspid and is substantially smaller than the distal teeth. A replacement pit is present on the
second tooth, but pits are present within dental tissues at other tooth positions that lack teeth. The
replacement pits are deep and almost circular. The teeth have substantial but not excessive
deposits of basal cementum.

The labial face of the dentary is moderately convex and has rugose sculpturing across
much of its surface, but rugosities are absent immediately ventral to the three distal teeth and on
the mesial-most portion of the element (Fig 2B). The coronoid facet is subtriangular and extends
anteriorly to the fourth tooth, and ventrally to just above the dorsoventral level of a row of
nutrient foramina. The facet is deep, moderately textured, and has a marked ventral boundary.
There are nine distinct nutrient foramina that extend distally to the second tooth, although most
of the foramina are concentrated mesially. The distal portion of the dentary is less complete than
the rest of the fossil, and so the posterior extent of the coronoid and angular processes is
uncertain, as is the presence of a separate surangular process. However, the coronoid process
does not appear to have a large dorsal extent.

45 46 47 **Diagnosis**

The presence of pleurodont teeth that are superficially attached to the lingual surface of the jaw
diagnoses YPM 4607 as a member of Lepidosauria (Estes et al. 1988; Gauthier et al. 1988).
Rhynchocephalians generally have acrodont teeth, although *Gephyrosaurus* has more apically
positioned pleurodont teeth (Evans 1980).

Asymmetrically bicuspid distal teeth and an open Meckelian groove are characteristic of
Teiidae, but are also present in Lacertidae, which is closely related to Teiidae.

Gymnophthalmidae + Alopoglossidae, the sister clade of Teiidae, can also have bicuspid teeth,
but the Meckelian groove is generally closed and fused (Bell et al. 2003). Amphisbaenia, the
sister taxon of Lacertidae, possesses several unique derived features, including unicuspid teeth, a
short dentary, a low tooth count (<10 teeth), and acrodont, subacrodont, or subpleurodont
dentition (Gans et al. 2008; Longrich et al. 2015). YPM 4607 has substantial deposits of
cementum at its tooth bases, large subcircular tooth replacement pits, and an elongate Meckelian
groove providing space for a hypertrophied splenial, all of which are apomorphies that diagnose
Teiidae to the exclusion of other squamates (Estes et al. 1988, Nydam et al. 2007).

Borioteioids can have all three dentary apomorphies commonly used to diagnose teiid
fossils (Nydam et al. 2007, see above) that are present in YPM 4607. The fossil is excluded from
Polyglyphanodontia because it lacks posterior teeth with transversely-oriented cusps, from
*Chamops* because the Meckelian groove is anteriorly restricted, there are no distinctly tricuspid
teeth, and the distal teeth are markedly less bulbous (Estes 1983), from *Prototeius* because the
posterior teeth are bicuspid and are not massive or blunt-cusped and the Meckelian groove is
anteriorly restricted (Denton and O'Neill 1995), and from members of Macrocephalosauridae,
which have flared, multicuspid teeth, sometimes with obliquely-oriented cusps (Sulimski 1975).
YPM 4607 is part of total clade Teiidae *sensu stricto* (excluding Borioteiodea).

The mesial restriction of the Meckelian groove excludes the fossil from Tupinambinae,
whose members have a completely unrestricted and medially facing Meckelian groove (Denton
and O'Neill 1995; Albino et al. 2013), but is a feature shared with Teiinae (Fig 3). Tupinambines
are also distinguished from the fossil by possessing a massive incision in between the coronoid
and angular processes of the dentary. The teiines *Dicrodon* and *Teius* have more basal cementum
relative to cnemidophorines and YPM 4607 (Fig 3G; Albino et al. 2013). The fossil also lacks
the transverse tooth cusps of *Dicrodon* (Fig 3G, 4G, 5G) and *Teius* (Presch 1974).

Among examined teiids (a complete list of specimens is in ESM 1), only *Aspidoscelis*
and *Holcosus* have sculpturing on the labial surface of the dentary (Fig 4A-C). Labial sculpturing
was reported to occur in adult individuals of *Ameiva* and *Cnemidophorus* without reference to
any particular species (Norell 1989) and before those genera were split up. Analyses by Tucker
et al. (2016) and Tucker et al. (2017) of an anchored phylogenomic dataset inferred *Aspidoscelis*
and *Holcosus* as sister taxa. Given that hypothesis, sculpturing on the dentary appears to be an
apomorphy of the *Aspidoscelis* + *Holcosus* clade. *Aspidoscelis* + *Pholidoscelis* were estimated as
sister taxa and *Holcosus* was sister to that clade in the analysis of Zheng and Wiens (2016), but
those relationships were supported by negligible bootstrap values. That relationship would imply
a single origin for sculpturing for the entire *Aspidoscelis* + *Pholidoscelis* + *Holcosus* clade or
independent origins in *Aspidoscelis* and *Holcosus* with a loss in *Pholidoscelis*. Alternatively,
*Aspidoscelis*, *Holcosus*, and *Pholidoscelis* were paraphyletic with respect to each other within
Teiinae in the analyses of Goicoechea et al. (2016). I accept that *Aspidoscelis*, *Holcosus*, and
*Pholidoscelis* form a clade to the exclusion of other extant teiid lizards, following Tucker et al.
(2016), Tucker et al. (2017), and Zheng and Wiens (2016).

Most specimens of *Aspidoscelis* lack sculpturing. In both *Aspidoscelis* and *Holcosus*,
sculpturing occurs often but not universally on larger and more robust specimens that were
probably skeletally mature adults, as was observed by Norell (1989). *Holcosus festiva* MVZ
79608 is a larger specimen that lacks any sculpturing. Sculpturing is present on all examined
specimens of *Aspidoscelis deppii*, some specimens of *Aspidoscelis tigris* (e.g., TxVP M-8629,
M-8631, M-15034), *Aspidoscelis gularis* TxVP M-15028, and on *Holcosus quadrilineatus* and
*Holcosus undulatus* UF 51244. In *Holcosus*, labial sculpturing ranges from longitudinal and

wispy texturing (*Holcosus undulatus*; Fig 4C) to more pronounced and vermiculate sculpturing
(*Holcosus quadrilineatus* UF 37170). In *Aspidoscelis*, labial sculpturing can be rugose
(*Aspidoscelis tigris*; Fig 4B) or less complexly textured (*Aspidoscelis deppii*; Fig. 4A). The
sculpturing of YPM 4607 is most similar to examined specimens of *Aspidoscelis tigris*. I did not
find sculpturing on specimens of *Ameiva*, *Cnemidophorus*, *Kentropyx*, *Medopheos*, or
*Pholidoscelis*. I sampled all cnemidophorine genera, but sampling of more species, particularly
of *Pholidoscelis*, is desirable to further establish the distribution of sculpturing within
cnemidophorines. I observed no sculpturing in tupinambines, *Dicrodon*, or *Teius*, and I did not
examine the teiines *Aurivela*, *Contomastix*, *Glaucomastix*, or *Ameivula*.

While tooth cusp morphology is known to vary ontogenetically and interspecifically in
teiids (Estes and Williams 1984), all examined specimens of *Holcosus*, *Ameiva*, *Cnemidophorus*,
*Kentropyx*, and *Medopheos* have exclusively tricuspid distal dentary teeth, as do *Aurivela*,
*Contomastix*, and *Ameivula* (see Estes 1963, Harvey et al. 2012, Presch 1974). In *Holcosus*,
*Ameiva*, *Cnemidophorus*, *Kentropyx*, and *Medopheos*, the transition from unicuspid teeth to
tricuspid teeth occurs on the mesial half of the tooth row, often abruptly. Many *Aspidoscelis* have
bicuspid teeth throughout the tooth row, but some species, such as *Aspidoscelis uniparens* and
*Aspidoscelis sonora*, have mostly tricuspid teeth. Some small specimens of *Pholidoscelis* have
mostly tricuspid distal teeth (e.g., *Pholidoscelis chrysolaeus* UF 99352), but in larger
specimens distal teeth are all bicuspid (Fig 3D, 4D) or are mostly bicuspid except for the teeth at
the last two or three tooth positions, which are tricuspid (e.g., *Pholidoscelis chrysolaeus* UF
99646). Having bicuspid distal teeth is either a separately derived state in *Aspidoscelis* and
*Pholidoscelis* with respect to other cnemidophorines, or is derived in the clade *Aspidoscelis* +
*Holcosus* + *Pholidoscelis*, with an apparent reversal in *Holcosus*. The third tooth of YPM 4607 is
sloped ventrodistally to create a slight dorsal shoulder on the distal face of the tooth. That
morphology also occurs in some specimens of *Aspidoscelis* that have bicuspid distal teeth (e.g.,
*Aspidoscelis deppii*, Fig 3A, 4A). I interpret the third tooth of YPM 4607 as representing a
shoulder instead of a third cusp, but recognize that there is some ambiguity.

The morphology of the Meckelian groove, labial sculpturing, and the presence of
bicuspid distal teeth place YPM 4607 in total clade *Aspidoscelis* + *Holcosus* + *Pholidoscelis*.
Based on the available material, labial sculpturing is unique to *Aspidoscelis* and *Holcosus*;
however, given that I examined only four of the twenty species of *Pholidoscelis*, I refrain from
excluding that clade for the time being and instead refer the fossil to total clade *Aspidoscelis* +
*Holcosus* + *Pholidoscelis*.

41 42 **Remarks**

YPM 4607 is unusual among teiids in having a strongly and abruptly tapered anterior portion of
the dentary. No examined teiids have a comparable morphology, although I observed moderately
stepped tapering of the dentary in *Cnemidophorus lemniscatus* (Fig 3F, 4F), *Aspidoscelis deppii*
(Fig 3A, 4A), *Aspidoscelis tigris* (e.g., TxVP M-15034, M-15035), and *Holcosus undulatus* (Fig
3C, 4C). Lateral **eversion** of the labial surface of the dentary is most pronounced in *Holcosus*
among extant teiids (Fig 5C). Correspondingly, the ventral border of the coronoid facet is most
distinct in *Holcosus*. The facet has a less defined ventral border in *Aspidoscelis*, *Pholidoscelis*,
and other cnemidophorines, and eversion of the dentary is less exaggerated in those taxa as well.
**Eversion** of the dentary is comparable between YPM 4607 and *Holcosus*.

Two of the four examined species of *Holcosus* had dentary sculpturing. The unusually complex and incompletely resolved phylogeny of *Aspidoscelis* (Barley et al. 2019; Reeder et al. 2002) and intraspecific and intra-clade variability in dentary sculpturing makes the evolution of that morphology difficult to interpret within *Aspidoscelis*. Some specimens of *Aspidoscelis tigris* have highly reduced sculpturing, including larger specimens (e.g., TxVP M-8630). *Aspidoscelis hyperthyrus*, which is closely related to *Aspidoscelis deppii*, lacks sculpturing altogether.

The presence of a moderately developed distal extension of the intramandibular septum (IMS) was reported to diagnose polyglyphanodontids to the exclusion of extant teiids (Denton and O'Neill 1995). However, I observed a well-developed distal IMS in several extant teiids, including *Aspidoscelis* and *Holcosus* (Fig 3A-C).

Discussion

To my knowledge, YPM 4607 is the oldest published record of a crown cnemidophorine from North America identified explicitly using apomorphies. The minimum age of total clade *Aspidoscelis* + *Holcosus* + *Pholidoscelis* and the minimum age of the crown cnemidophorine clade is 6.6 ± 0.3 Ma, based on a fission track age estimate of glass from the Ash Hollow Formation (Skinner and Johnson 1984). Thus, 6.3 Ma can be used as a minimum age for the divergence between the *Aspidoscelis* + *Holcosus* + *Pholidoscelis* clade and the *Ameiva* + *Cnemidophorus* + *Kentropyx* + *Medopheos* clade (i.e., the age of crown cnemidophorines). Few fossils were previously considered reliable for bracketing minimum clade ages in divergence time analyses of Teiidae, and only one cnemidophorine of indeterminate phylogenetic affinity was used recently (Albino et al. 2013; Tucker et al. 2017). YPM 4607 is significant for future studies seeking to temporally calibrate the evolutionary history of teiids and cnemidophorines in particular.

The resurrection of the genera *Aspidoscelis*, *Holcosus*, and *Pholidoscelis* for clades previously accommodated in *Ameiva* and *Cnemidophorus* (Goicoechea et al. 2016; Harvey et al. 2012; Reeder et al. 2002) was helpful here for clarifying the systematic significance of two morphologies in cnemidophorines. Labial sculpturing of the dentary was reported in adult specimens of *Ameiva* and *Cnemidophorus* by Norell (1989). Given the sample here, sculpturing is restricted to two clades formerly in *Ameiva* and *Cnemidophorus*, *Holcosus* and *Aspidoscelis*, that are hypothesized to be each other's closest relatives (Tucker et al. 2016; 2017). Similarly, bicuspid distal tooth crowns were previously reported in certain groups of *Ameiva* and *Cnemidophorus* (Presch 1974; Estes 1963). The West Indian *Ameiva* (i.e. *Pholidoscelis*) were recognized as the only *Ameiva* to have bicuspid teeth by Estes (1963), but no pattern was perceived in *Cnemidophorus*. *Pholidoscelis* and *Aspidoscelis*, which are closely related, are the only genera previously assigned to *Ameiva* and *Cnemidophorus* that have bicuspid distal teeth. Tooth cusp morphology has been an enigmatic feature to interpret in cnemidophorine teiids, but now appears to be elucidated systematically. The presence of dentary sculpturing and bicuspid teeth are apomorphies that can be used to identify cnemidophorine fossils in the future, and the presence or absence of those features could potentially be used as characters in phylogenetic analyses of morphological data.

Aspidoscelis sexlineatus is the only teiid that occurs in Nebraska or immediately adjacent to Nebraska in the modern biota (Stebbins 2003), indicating that a different lineage of cnemidophorine was found in the area at least as recently as the middle-late Miocene. *Aspidoscelis tigris*, the *Aspidoscelis tessellatus* complex, the *Aspidoscelis neotesselatus* complex,

and *Aspidoscelis velox* are currently found in southern and western Colorado (Stebbins 2003).
*Pholidoscelis* is currently restricted to islands in the Caribbean, and *Holcosus* is found in Central
and South America and in Mexico as far north as central Tamaulipas (Goicoechea et al. 2016;
Lavín-Murcio and Lazcano 2010). More fossils from across North America are needed to
explore the historical biogeography of cnemidophorine lizards and to determine whether modern
tropical or island clades like *Holcosus* and *Pholidoscelis* were once found farther north on the
mainland or on the mainland at all, respectively.

The present survey of extant teiids indicates that the overall morphotype of YPM 4607 is
unique, particularly the mesial tapering of the dentary. No teiid taxon found in the extant biota
has a directly comparable set of features. However, the fossil cannot be definitively excluded
from total or crown *Aspidoscelis*, *Pholidoscelis*, or *Holcosus*, so for now I refrain from
establishing a new taxon should new material reveal that YPM 4607 is referable to one of those
clades. Additional fossils of the lineage to which YPM 4607 belonged are needed to refine its
systematic position, investigate whether other skeletal elements were similarly distinctive, and
determine whether a new taxon is warranted.

**Ethics**

No permissions were required. YPM 4607 is deposited at the Yale Peabody Museum.

**Data accessibility**

Electronic supplementary material is available as supplementary file ESM 1 at *Royal Society*
*Open Science*.

**Competing interests**

The author declares no competing interests.

**Funding**

[revised manuscript text omitted]

Chovanec K. 2014 Non-anguimorph lizards of the late Oligocene and early Miocene of Florida
and implications for the reorganization of the North American herpetofauna. M.Sc. Thesis,
Department of Geosciences, East Tennessee State University. 123 pp. Available from
<https://dc.etsu.edu/cgi/viewcontent.cgi?article=3732&context=etd>. Accessed 23 June 2016.

Cope ED. 1862 Synopsis of the species of *Holcosus* and *Ameiva*, with diagnoses of new West
Indian and South American Colubridæ. *Proceedings of the Academy of Natural Sciences of*
*Philadelphia* **14**, 60–82.

Daza JD, Stanley EL, Wagner P, Bauer AM, Grimaldi, DA. 2016 Mid-Cretaceous amber fossils
illuminate the past diversity of tropical lizards. *Science Advances* **2**, e1501080.
<https://doi.org/10.1126/sciadv.1501080>

Denton Jr RK, O'Neill RC. 1995 *Prototeius stageri*, Gen. et sp. Nov., a new Teiid lizard from
the Upper Cretaceous Marshalltown Formation of New Jersey, with a preliminary phylogenetic
revision of the Teiidae. *Journal of Vertebrate Paleontology* **15**, 235–253.

Estes R. 1963 Early Miocene salamanders and lizards from Florida. *Quarterly Journal of Florida*
*Academy of Sciences* **25**, 234–256.

Estes R. 1983 *Encyclopedia of Paleoherpetology, Sauria terrestria, Amphisbaenia*. Gustav Fisher
Verlag, Stuttgart, Germany, xxii+249 pp.

Estes R, Tihen JA. 1964 Lower vertebrates from the Valentine Formation of Nebraska. *The*
*American Midland Naturalist* **72**, 453–472.

Estes R, Williams EE. 1984 Ontogenetic variation in the molariform teeth of lizards. *Journal of*
*Vertebrate Paleontology* **4**, 96–107.

Estes R, De Queiroz K, Gauthier J. 1988 Phylogenetic relationships within Squamata. In
Phylogenetic Relationships of the Lizard Families: Essays Commemorating Charles L. Camp
(pp. 119–281). Stanford, California: Stanford University Press.

Evans SE. 1980 The skull of a new eosuchian reptile from the Lower Jurassic of South Wales.
Zoological Journal of the Linnean Society **70**, 203–264. [https://doi.org/10.1111/j.1096-](https://doi.org/10.1111/j.1096-3642.1980.tb00852.x)
[3642.1980.tb00852.x](https://doi.org/10.1111/j.1096-3642.1980.tb00852.x)

Fitzinger L. 1843 Systema Reptilium, fasciculus primus, Amblyglossae. Braumüller et Seidel,
Wien: 106 pp.

Gans C, Montero R. 2008 An atlas of amphisbaenian skull anatomy. In *Biology of the Reptilia,*
*Volume 21* (eds C Gans, AS Gaunt, K Adler), pp. 621–738. Ithaca, NY: Society for the Study of
Amphibians and Reptiles.

Gauthier J, Estes R, de Queiroz K. 1988 A phylogenetic analysis of Lepidosauromorpha. In
*Phylogenetic Relationships of the Lizard Families: Essays Commemorating Charles L. Camp*
(eds R Estes, GK Pregill), pp. 15–98. Stanford, CA: Stanford University Press.

Gauthier JA, Kearney M, Maisano JA, Rieppel O, Behlke ADB. 2012 Assembling the squamate
tree of life: Perspectives from the phenotype and the fossil record. *Bulletin of the Peabody*
*Museum of Natural History* **53**, 3–308. <https://doi.org/10.3374/014.053.0101>

Goicoechea N, Frost DR, Riva I, Pellegrino KCM, Sites JJ, Rodrigues MT, Padiá JM. 2016
Molecular systematics of teioid lizards (Teioidea/Gymnophthalmoidea: Squamata) based on the
analysis of 48 loci. *Cladistics* **32**, 1–48. <https://doi.org/https://doi.org/10.1111/cla.12150>

Gray JE. 1827 A synopsis of the genera of Saurian reptiles, in which some new genera are
indicated, and the others reviewed by actual examination. *The Philosophical Magazine* **2**, 54–58.
<https://doi.org/10.1080/14786442708675620>

Harvey MB, Ugueto GN, Gutberlet RL. 2012 Review of teiid morphology with a revised
taxonomy and phylogeny of the Teiidae (Lepidosauria: Squamata). *Zootaxa* **3459**, 1–156.

Holman JA. 1975 Herpetofauna of the WaKeeney Local Fauna (Lower Pliocene: Clarendonian)
of Trego County, Kansas. *University of Michigan Papers in Paleontology* **3**, 49–66.

Joeckel RM. 1988 A new late Miocene herpetofauna from Franklin County, Nebraska. *Copeia*
**1988**, 787–789.

Lavín-Murcio PA, Lazcano DA. 2010 Geographic distribution and conservation of the
herpetofauna of Northern Mexico. In *Conservation of Mesoamerican reptiles and amphibians*
(eds LD Wilson, JH Townsend, JD Johnson), pp. 274–301. Eagle Mountain, UT: Eagle Mountain
Publishing, LC.

Lee MSY. 2009 Hidden support from unpromising data sets strongly unites snakes with
anguimorph “lizards.” *Journal of Evolutionary Biology* **22**, 1308–1316.
<https://doi.org/10.1111/j.1420-9101.2009.01751.x>
Longrich NR, Vinther J, Pyron RA, Pisani D, Gauthier, JA. 2015 Biogeography of worm lizards
(Amphisbaenia) driven by end-Cretaceous mass extinction. *Proceedings of the Royal Society B*
**282**, 1–10. <https://doi.org/10.1098/rspb.2014.3034>
Norell MA. 1989 Late Cenozoic lizards of the Anza Borrego Desert, California. *Contributions in*
*Science, Natural History Museum of Los Angeles County* **414**, 1–31.
Norell, MA, Queiroz K De. 1991 The earliest iguanine lizard (Reptilia: Squamata) and its
bearing on iguanine phylogeny. *American Museum Novitates* **2997**, 1–16.
Nydam ARL, Eaton JG, Sankey J. 2007 New taxa of transversely-toothed lizards (Squamata:
Scincomorpha) and new information on the evolutionary history of “teiids.” *Journal of*
*Paleontology* **81**, 538–549.
Parmley D, Peck D. 2002 Amphibians and reptiles of the late Hemphillian White Cone local
fauna, Navajo County, Arizona. *Journal of Vertebrate Paleontology* **22**, 175–178.
[https://doi.org/10.1671/0272-4634\(2002\)022\[0175:AAROTL\]2.0.CO;2](https://doi.org/10.1671/0272-4634(2002)022[0175:AAROTL]2.0.CO;2)
Parmley D, Bahn, JR. 2012 Late Pleistocene lizards from Fowlkes Cave, Culberson County,
Texas. *The Southwestern Naturalist* **57**, 435–441. <https://doi.org/10.1894/0038-4909-57.4.435>
Perkins ME, Nash BP. 2002 Explosive silicic volcanism of the Yellowstone hotspot: The ash fall
tuff record. *Bulletin of the Geological Society of America* **114**, 367–381.
[https://doi.org/10.1130/0016-7606\(2002\)114<0367:ESVOTY>2.0.CO;2](https://doi.org/10.1130/0016-7606(2002)114<0367:ESVOTY>2.0.CO;2)
Presch W. 1974 A survey of the dentition of the macroteiid lizards (Teiidae: Lacertilia).
*Herpetologica* **30**, 344–349.
Pyron RA. 2017 Novel approaches for phylogenetic inference from morphological data and
total-evidence dating in squamate reptiles (lizards, snakes, and amphisbaenians). *Systematic*
*Biology* **66**, 38–56. <https://doi.org/10.1093/sysbio/syw068>
Reeder TW, Cole CJ., Dessauer HC. 2002 Phylogenetic relationships of whiptail lizards of the
genus *Cnemidophorus* (Squamata: Teiidae): A test of monophyly, reevaluation of karyotypic
evolution, and review of hybrid origins. *American Museum Novitates* **3365**, 1–61.
[https://doi.org/10.1206/0003-0082\(2002\)365<0001:prowlo>2.0.co;2](https://doi.org/10.1206/0003-0082(2002)365<0001:prowlo>2.0.co;2)
Reeder TW, Townsend TM, Mulcahy DG, Noonan BP, Wood Jr PL, Sites Jr JW, Wiens JJ. 2015
Integrated analyses resolve conflicts over squamate reptile phylogeny and reveal unexpected
placements for fossil taxa. *PLoS ONE* **10**, 1–22. <http://dx.doi.org/10.1371/journal.pone.0118199>

RStudio Team. 2015 RStudio: Integrated Development for R. RStudio, Inc., Boston, MA.
<http://www.rstudio.com/>

Simões TR, Caldwell MW, Talanda M, Bernardi M, Palci A, Vernygora O, Bernardini F,
Mancini L, Nydam RL. 2018. The origin of squamates revealed by a Middle Triassic lizard from
the Italian Alps. *Nature* **557**, 706–709. <https://doi.org/10.1038/s41586-018-0093-3>

Skinner M, Johnson F. 1984 Tertiary stratigraphy and the Frick Collection of fossil vertebrates
from north-central Nebraska. *Bulletin of the American Museum of Natural History* **178**, i + 217-
368.

Stebbins RC. 2003 A field guide to western reptiles and amphibians. 3rd ed. New York, NY:
Houghton Mifflin Company.

Sulimski A. 1975 Macrocephalosauridae and Polyglyphanodontidae (Sauria) from the Late
Cretaceous of Mongolia. *Palaeontologia Polonica* **33**, i + 26-102 + plates VIII-XXVII.

Swisher CC. 1992 ⁴⁰Ar/³⁹Ar dating and its application to the calibration of the North American
land mammal ages. PhD Dissertation, University of California, Berkeley. 239 pp.

Taylor EH. 1941 Extinct lizards from upper Pliocene deposits of Kansas. *State Geological*
*Survey of Kansas Bulletin* **38**, 165–176.

Tedford RH, Alrbright LBA, Barnosky AD, Ferrusquia-Villafranca I, Hunt RMJ, Storer JE,
Swisher CC, Voorhies MR, Webb SD, Whistler DP. 2004 Mammalian biochronology of the
Arikareean through Hemphillian interval (late Oligocene through early Pliocene epochs). In *Late*
*Cretaceous and Cenozoic Mammals of North America: Biostratigraphy and Geochronology* (ed
MO Woodburne), pp. 169-231. New York, NY: Columbia University Press.

Tennekes M. 2018 “tmap: Thematic Maps in R.” *Journal of Statistical Software* **84**, 1–39.
<http://dx.doi.org/10.18637/jss.v084.i06>

[revised manuscript text omitted]

100x168mm (300 x 300 DPI)

Appendix C

Hi all,

Thank you for the thorough reviews and critiques of the manuscript, I appreciate it. I have revised the manuscript considerably, and I hope you all agree that it is improved. Most of the suggested references and suggestions have been integrated into the text, and several parts of the manuscript have been substantially reorganized. All comments are listed below and addressed individually, explaining how I either revised the text, or why I disagree.

I also want to mention that, with respect to the specimen number of the fossil, 4607 was a typo that I repeated throughout the manuscript. The specimen number is actually 4707, and I have modified the text accordingly. Sorry about that!

Reviewer 1

Dear Author,

Your paper constitutes, in my opinion, an interesting discovery that will for sure help to better understand the history of this poorly known (from a paleontological and osteological point of view) group of lizards.

I am not a specialist of North American squamates so I will not be able to provide much advice regarding most of the taxa discussed in your paper but I however did some work on the osteology of *Pholidoscelis*. As such, I believe that more precision could easily be added in the paper regarding the dentary and teeth morphology of this genus. All my comments regarding this are based on a work recently published regarding the fossil *Pholidoscelis* of Guadeloupe in the framework of which I wrote a short synthesis regarding the teeth morphology and osteological variability of modern *Pholidoscelis*. You seem to have missed this publication (Bochaton et al., 2019) which could maybe help you to bring more precision to the following points:

Thank you for your detailed comments. I particularly appreciate your thoughts on the morphology of *Pholidoscelis*, and I agree that that aspect of the manuscript was lacking.

-You seem to have missed this publication (Bochaton et al., 2019) which could maybe help you to bring more precision to the following points:

Thank you for bringing this publication to my attention!

-I think your description of the ontogenetic evolution of the teeth of *Pholidoscelis* lacks precision and reference to the appropriate literature. This variability was first described by Pregill (Pregill, 1981; Pregill et al., 1988) and I tried to expand the description with specimens of most of the species (Bochaton et al., 2019). The general ontogenetic morphological evolution regarding the posterior teeth is tricuspid->bicuspid->monocuspid->bulbous in Lesser Antillean species. In the Greater Antilles most species “stop” this evolution before reaching the last steps and preserve tricuspid or bicuspoid posterior teeth in large/adult specimens. This is however not a strict rule (see my comment on *Ameiva*).

Added this discussion to the text, thanks again for pointing this out.

-Pholidoscelis dentaries of large specimens do present some dermal ornamentation. Not as on your fossil through but limited to the anterior part of the bone (Bochaton et al., 2019).

This is good to know and I have modified the text to reflect these observations. This makes the diagnosis more straightforward.

-Some large specimens of Ameiva present bicuspid posterior/distal teeth. This was observed on a specimen from the MNHN, Paris (Bochaton et al., 2019).

This is also very useful, and does modify the diagnosis somewhat (see expanded diagnosis).

-The number of teeth of your specimen (19) falls into the variability of Pholidoscelis (15 to 27) (Bochaton et al., 2019). A comparison between the length and the dental row (absent in the paper?) and the number of teeth would maybe help to discard an attribution to Pholidoscelis?

It wasn't my intent to eliminate *Pholidoscelis*. I understand that such a comparison may be useful in the Holocene or Pleistocene when you are more likely to encounter extant species, but I am not sure what purpose it would serve here.

-These comments led to the conclusion that the fossil described in this study cannot be attributed to Pholidoscelis, considering that the pattern of dermal sculpturing present on the fossil was previously stated in the literature as absent in this genus. When I first saw the fossil I have been impressed by the posterior height of its labial margin which is also high in large Pholidoscelis (there is ontogenetic variability in this). But your fossil seems very small (10.5mm of dental length following the picture) compared to the member of Pholidoscelis in which a similar morphology occurs (20-30mm of dental row length).

I think you have misinterpreted or misread this aspect of the diagnosis. While I did not personally observe any specimens of *Pholidoscelis* with labial sculpturing, I intentionally did not exclude *Pholisoscelis*, and left open the possibility that other specimens of *Pholisoscelis* that I did not observe might have sculpturing (and after reading your paper, I now know that they do):

“Based on the available material, labial sculpturing is unique to *Aspidoscelis* and *Holcosus*; however, given that I examined only four of the twenty species of *Pholidoscelis*, I refrain from excluding that clade for the time being and instead refer the fossil to total clade *Aspidoscelis* + *Holcosus* + *Pholidoscelis*.” (p. 6 l. 34-40)

That aspect of the diagnosis has been revised. I don't think you can make any conclusions based on the size of the fossil; it is at from at least 6 mya and body size can change dramatically over much shorter time scales. For what it's worth, the fossil is the same size as many adult *Aspidoscelis* that have sculpturing.

-Another very important point regarding your interpretation of the fossil: in regard to lesser Antillean “Ameiva” (=Pholidoscelis), Hedges (2006) indicates a divergence time of

45Ma, so much older than your fossil. This should be discussed as it is in strong contradiction with your interpretation.

Again, I think you have misinterpreted what I said (or maybe I was not clear?). There is nothing contradictory about older divergence times relative to a first known fossil occurrence. That is an expected and well-documented phenomenon—although that incongruence may lead to discrepant phylogenetic and biogeographic hypotheses. I did not say that this is the oldest fossil cnemidophorine from North America that has ever been described, or that it is temporally adjacent to the beginning of the crown or total clade. I said that it is the oldest known fossil of a crown cnemidophorine from North America that was identified using apomorphies, and that it can be used to calibrate divergence time analyses. Also, there are better divergence time estimates based on a more appropriate and larger dataset (Tucker et al. 2017) than that of Hedges (2006).

What I said was almost the opposite—it is remarkable that the record of clearly identified fossil cnemidophorines is so sparse in North America during the Neogene (and non-existent in the late Paleogene) given published divergence times.

The introduction seems incomplete and is mostly limited to contextual information. The question of the study is not clearly stated here. What is the aim of publishing this fossil? Why was it studied? The introduction contains sufficient context regarding other finds of fossil teiids in the Americas but this part would be much better with a broader introduction at the start and a proper introduction of the research questions. Also regarding your references, could you split them regarding the different geographic areas, even in a single sentence? This would make easier to know which author speaks about what.

I have added some text in the introduction and the abstract to make the aim of publishing the fossil more clear- the fossil is useful as a node calibration, it is the oldest known crown cnemidophorine, and therefore is a step towards a better understanding of teiid evolution in North America. I disagree with several of your other points. In my opinion, it is important for scientists to sometimes report our observations without needing to have a specific question in mind, and that is how I approached this project.

I split up those references in the sentence discussing Quaternary fossil teiids.

Geological context

This part also lacks important information in my opinion. That would be great to see at least a rapid presentation of the other taxa found in that formation, squamates, or not. That would also help to better understand why this fossil is important. I discovered in the caption of figure 1 that this locality was previously studied as there is another reference for it? This should be clearly presented in this part. A reference to Fig. 1 should also be added.

Figure reference for Fig 1 is in there, but buried in a parenthetical comment. I put the reference in its own parentheses and added another Fig 1 reference. I thought I had been clear in the introduction that the area had been studied before (Estes and Tihen reference at end of introduction), but added a reference to Cherry County to make it more explicit.

The reason I did not go into more detail about the Valentine or Ash Hollow formations is because there is no way of knowing which formation the fossil was collected from. I added some information about mammal taxa in the biochronology section in Materials and Methods, which helped better constrain the fossil temporally.

Also, you keep referring to the fact that more material would be needed to identify this fossil but it is unclear whether this material exists/could exist or not. Could you clarify this?

There are two ideas being referenced here and I have not distinguished them clearly, sorry about that.

First (skeletal material)- I did not look at all *Pholidoscelis*. I intentionally did not exclude the fossil from *Pholidoscelis* for that reason, and I intended to recognize that if I looked at more species/specimens, my observations might change. You have helped clarify this issue (*Pholidoscelis* can have sculpturing).

Second (fossil material)- I do not think that YPM VP 4707 can itself be referred to *Aspidoscelis*, *Pholidoscelis*, or *Holcosus*, and it could conceivably be a member of a new clade related to some or all of those. Additional fossil material of the lineage to which YPM VP 4707 belonged will hopefully clarify the present uncertainty (last paragraph of discussion).

Description

Please add the usual measurements in the description, measuring it on the picture might be a source of error.

Added tooth row length

Reviewer 2

I have reviewed the manuscript “Unusual lizard fossil from the Miocene of Nebraska and a minimum age for cnemidophorine teiids” (RSOS-200317). The manuscript is original in describing a dentary from the Miocene of the USA, its assignment to the cnemidophorine, would make it the oldest fossil of the clade, and as the authors states it presents a minimum age for them, useful in age calibration studies.

The manuscript is well written (but see below) and accessible to a wide audience. The illustrations are very good and useful. The methodology and methods are appropriate, but I have some consideration on the conclusions that I believe should be addressed.

I have commented directly on the pdf, but my main observations are the following:

Thanks for your helpful critiques, and for catching the missed references and parts of the text that were confusing or convoluted. Much appreciated!

Evans 2008; Zheng and Wiens 2016

Thank you for catching my missed references! They are now added

"containing *Aspidoscelis*, *Holcosus* and *Pholidoscelis*."

With "+" readers could interpret as a polytomy

Changed to ‘containing *Aspidoscelis*, *Holcosus* and *Pholidoscelis*’

• I would recommend the author to improve some passages of the text that are complicated to follow (as the paragraph describing sculpturing in the different taxa). And in other passages it is not clear the phylogenetic hypothesis followed (p6 11, p7 136-26).

I have amended the text to be clear about which taxonomy and phylogenetic hypothesis I am following (Tucker et al. 2016; 2017).

• Regarding terminology I would suggest using distal, mesial, labial and lingual for teeth, and posterior, anterior, medial and lateral for the dentary throughout the entire text. As it is the reading of the manuscript is complicated (at least to me).

Reviewer 3 made the same comment about mesial and distal, should all be fixed to reflect your comment now. As for labial/lingual and medial/lateral I have heard both with respect to the dentary itself, and so have left those terms as is.

• In the description please check what is identified as intramandibular septum which seem to be the intramandibular lamella. Please see Denton and O’neill (1995), Lee and Scanlon (2001) and Smith (2009).

I have added some clarification on this point in the text (pointing out that the posterior portion is also called the intramandibular lamella).

there is a contradiction in writing (136 vs141), teiids are or not present in the North American Paleogene

I phrased that sentence confusingly. What I intended to say is that, given published divergence times, it is remarkable that there are no known teiids from the Paleogene of North America. Hopefully this is fixed now.

• In the description, the author describes implantation as pleurodont, but to me it is subpleurodont (as most other authors consider implantation in teiids). If authors disagrees it should be disused.

I decided to describe the dentition as simply ‘pleurodont’ because I have seen several (generally taxon-dependent) definitions and usages of ‘subpleurodont’, and current literature does not distinguish subcategories of pleurodony (see Bertin et al. 2018). I added a mention in the expanded diagnosis section that teiid teeth are often described as subpleurodont, but have left the description as ‘pleurodont’.

• Please see to consider the Barbatteiidae, a European Cretaceous clade considered sister to the extant teiids (Venczel and Codrea 2016; Codrea et al. 2017).

Thank you for bringing this clade to my attention! I have included Barbatteiidae in the diagnosis.

- **In the “Diagnosis” section (section should be re-titled).**

Personally, I do not like ‘Diagnosis’ sections that present a list of morphological features divorced from their systematic interpretation. I have tried to be specific in the new diagnosis section as to the meaning of each feature, and everything else is now in a section titled ‘Expanded Diagnosis and Discussion’.

if it is a shoulder than it would be a diagnostic character (I don't know of any Teiidae with bicuspid teeth and a distal shoulder). Alternative the mesial cusp could have its apex in a more central position than the other preserved teeth, hence the outline. I interpret a shoulder as a maked, discrete charcater, which I dont see (in the pdf image)

1) if the tooth presents a “shoulder” than it would be a diagnostic character (I don't know of any Teiidae with bicuspid teeth and a distal shoulder), but I interpret a shoulder as a defined and constant character on a given tooth. From the image this seem to be the mesial cusp with its apex in a more central position than the other preserved teeth, hence the outline.

I don't agree here. Other species of *Aspidoscelis* (one of which is illustrated, see *Aspidoscelis deppii*) have a similar shoulder/sloping morphology on the distal margin of some of the bicuspid teeth.

2) fossil YPM 4607 is lacking the posterior most teeth, which could or not be tricuspid. Presch (1974) states that some Ameiva and Cnemidophorus spp can have only one posterior tricuspid tooth. Therefore in l34-36 bicuspid posterior teeth cannot be consider as a characteristic of the clade.

True, although the *Ameiva* and *Cnemidophorus* of Presch (1974) included all of the other cnemidophorines besides *Kentropyx*. I have changed this section to be more specific, but in *Ameiva*, *Cnemidophorus*, *Kentropyx*, *Aurivela*, *Ameivula*, and *Medopheos* (as the genera are delimited in Tucker et al. 2017), all or almost all of the distal teeth are tricuspid, not just the distalmost tooth/teeth. Reviewer 1 pointed out one specimen that is an exception.

• In the “Remarks” section. I have noted that in teiids the "lateral eversion" is related to size (development of the adductor musculature)(don't recall if this is mentioned in any bibliography). Did the author compare to different size *Holcosus*? The specimen in Fig 4 and 5 (UF 51244) is the largest of those presented enforcing this hypothesis. I have doubts on its diagnostic use.

I did notice that the smaller *Holcosus* have less exaggerated expansion, but larger specimens of other genera did not have as expanded dentaries as did larger *Holcosus*. Regardless, I didn't actually use that feature to diagnose the fossil.

- **P8, l8 *Cnemidophorus* sp MCZ 3381 is from the lower Miocene Florida in (Estes 1963,**

1983:90). Both papers cited in manuscript, It would be older? Is so it would be the minimum age of the cnemidophorine clade?

See discussion with Reviewer 3. That fossil is certainly older, and in my opinion there is nothing about the fossil to suggest that it is not a cnemidophorine. The identification was not apomorphy-based, but more importantly, there is nothing in the original description or preserved on the fossil (as far as I can tell from the illustration) indicating that the fossil is a crown cnemidophorine. It is probably referable to the total clade, but I think that the identification should be formally reevaluated.

Reviewer 3

What a cool specimen! It is clearly an important contribution to the knowledge of true Teiidae in North America. The scoring in the review is not as bad as it appears, but is the result of insufficient categories and choices to more clearly indicate the areas of success and the areas in need of improvement. I like the work, but I have many well intentioned suggestions for improvement. The anatomical treatment is good, but the language is unnecessarily complicated (see my notes in the manuscript) and can be easily modified to be more accessible and clear. The author is missing some relevant literature such as the recent dental studies of Aaron LeBlanc and the Nydam et al., 2010 JVP article redefining Chamopsiidae. The reason for the request for Major Revision is due to the unfortunate lack of a meaningful exploration of the meaning of this fossil in the fossil record. As the purported oldest cnemidophorine specimen in North America (regardless of method of diagnosis) it provides evidence to address the hypotheses of the timing and mechanism of the arrival of true Teiidae in North America. The addition of such a discussion would dramatically increase the impact of this paper. Please see my comments in the manuscript and I wish the author the best of luck as he is clearly a very skilled burgeoning paleoherpetologist.

Glad you thought the specimen was cool as well, and thanks for the kind words. I appreciate your comments, which were very helpful for improving the manuscript.

I am familiar with using mesial and distal when describing teeth, but it is more common to use anterior and posterior when orienting to an element like the dentary.

See comments to Reviewer 2. I can understand how this was confusing, and I was not consistent in my use. Now changed to be consistent with the usage of distal/mesial that you both recommended.

Eversion is to turn the inside outward. I believe the condition illustrated here and common to many taxa to some degree is a lateral expansion.

Changed to expansion

I am confused. Does the author intend to say that there is an articulation facet for the splenial along the ventral margin of the posterior half of the "supramecklian lip" (another

relatively new term added unnecessarily to the anatomical venacular when subdental lamina is sufficient).

Changed to articulation facet.

Suprameckelian lip may be a relatively new term, but subdental lamina is hardly the only other term that is commonly used to describe that morphological feature (subdental ridge, subdental shelf, upper border of the Meckelian canal, I'm sure there are more). I think suprameckelian lip is more descriptive and I provided a reference for the term, so I don't think it's an issue.

The intramandibular septum is a valuable taxonomic and systematic feature. The description here does not give sufficient information about the morphology of the IMS (i.e., anteroposteriorly short, dorsoventrally short). The anatomical description should paint a picture of the element sufficiently clear that it can be used in absence of the element as well as to be sure that the author is properly communicating the anatomy in question. This is not a question of honesty, but a question of dispassionate observation.

Added description for ims

"Moderately well-developed" seems too subjective. I recommend actually describing the anatomy of its depth along length, width along length, and any other features (e.g., interruption by tooth replacement processes, condition as symphysis, etc.).

Agreed. I have changed that part to be more descriptive.

The bases are pretty closely spaced so I am assuming the author means that the crowns are "somewhat" widely spaced. Is this spacing or is it a function of the tapering of the tooth crowns towards their apices? Based on what I can observe from the images provided the bases are robust and the crowns become mesio-distally narrower towards the apex. The "mesiodistally expanded bases of the posteriormost tooth is consistnet with the molarization of this part of the tooth row in many taxa, but particularly in teiids.

This seems like a subjective disagreement about what is close or far-spaced. To me, the bases are neither close-spaced nor widely spaced, but I changed the description to specify the bases and be more specific. Also, I had already made it clear in the text that the bases are wider than the crowns.

I understand the use of the level of the nutrient foramina, but this reads as if the labial coronoid facet extends to the foramina or that they extend posteriorly to the facet. I think it would be more accurate to describe the ventral extent of the coronoid facet relative to a more closely neighboring feature such as the lateralmost extent of the lateral expansion of the posterior portion of the dentary.

Agreed that that was confusing. Changed to "The coronoid facet is subtriangular and extends anteriorly to the fourth tooth, and ventrally to the ventrolateral extent of the posterolateral expansion of the dentary."

This appears to be contradictory. I am uncertain how the author can say that the element is incomplete such that the posterior extent of the coronoid process is unknown, but then say its dorsal extend is possibly known when it is the same incomplete feature. I don't disagree with the possibility, but based on the preservation it must be uncertain and absent a value judgment in the description.

Removed the second sentence

I strongly recommend a thorough reading of the recent works of LeBlanc in regards to the anatomically/histologically updated assessments of tooth attachment definitions.

As far as I can tell there was only one terminological problem, which was fixed by changing “pleurodont teeth that are superficially attached to the lingual surface of the jaw” (the way it was described historically) to “pleurodont tooth implantation.”

Additionally, since this paper is purported be apomorphy-based I wonder if the Estes et al and Gauthier et al references are not outdated (see the references in the introduction) based on the more recent treatments of lepidosauromorpha systematics?

I am confused by “purported”- does this paper not present an apomorphy-based diagnosis?

If I was doing a phylogenetic analysis, I would use a more recently published matrix than Estes et al (1988) or Gauthier et al. (1988). Our knowledge has certainly increased in terms of the quantity of known morphological data, and our capacity to describe and frame those data as characters for phylogenetic analysis. Attention to the latter point is a particularly meaningful contribution of Simoes et al. (2018). At the level of individual characters, I completely disagree—Simoes et al. (2018) publishing a more recent treatment of the phylogeny of lepidosauromorphs does not somehow make characters described in older publications “outdated.” The only issue here, pleurodont tooth implantation nomenclature, was easily remedied. The most important aspect of framing morphological character evolution is the choice of phylogenic hypothesis used to interpret the characters, not how recently the characters in question were published or how the original authors interpreted the evolution of those characters with a different topology in mind. I added Simões et al. (2018) to that reference list.

While Chamops is certainly one of the most recognizable of Chamopsiidae (sensu Nydam et al., 2010) from the Late Cretaceous of North America. There are other taxa as well and a properly complete differential should include these taxa. At the very least this specimens can be excluded from Chamopsiidae by its lack of a pronounced symphyseal boss on the dentary and presence of an intramandibular septum.

I appreciate the perspective on other chamopsiids. I have added that to the diagnosis.

Indeed. In my experience of looking all of the specimens I could find of these taxa at several institutions led me to conclude that teeth are reliably consistent at the genus level for overall pattern, but as would be expected in a polyphyodont system there is substantial variation to be found from individual to individual. [distal shoulder of tooth]

Agreed.

I agree with the first part of the sentence, but the second part is simply a reflection of the timing of the manuscript development. I find it to be inappropriate to identify the technique as part of the discovery. This habit of aggrandizement has become far too common in the literature and if the author feels that strongly then he should point out all of the errors in the papers published before 2012 when the shift to apomorphy-based definitions began to overwhelm differential diagnosis. Quite simply, the specimen is the oldest known chemidophorine (crown or not), period. The technique does not change what it is, only how it was evaluated.

The perspective that you presented is flawed and misrepresents my views on this issue. First, I don't recall saying that all non-apomorphy based fossil identifications are in error. Adoption of an apomorphy-based identification system is not an explicit refutation of identifications made with other methods. Fossils that are not identified with apomorphies or phylogenetic analysis have limited use in secondary works (e.g., divergence time analyses), and using an apomorphy-based diagnosis ameliorates the types of biases and errors than can occur when using other identification criteria (such as phenetic resemblance or modern biogeography). In the case of cnemidophorines, the primary factors that have historically hindered fossil identification are problematic taxonomy and a record composed of fragmentary fossils (see Introduction). Also, the suggestion that 2012 is a general turning point is an overstatement; Quaternary paleontologists have yet to widely adopt apomorphy-based diagnoses.

Second, identifications can and do vary among apomorphy-based diagnoses. Using apomorphies requires reference to a specific tree topology or set of tree topologies with which the evolution of morphological features is interpreted. I accepted a hypothesis inferred by analyses of targeted sequence capture data. Although the fossil would still be identified as a cnemidophorine if I accepted the hypothesis of Goicoechea et al. (2016), its placement within the clade would be different. That aspect of the methodology is central to understanding the discovery, and to decontextualize the finding from the framework in which it was interpreted would be misleading.

Third, regarding your last comment (“Quite simply, the specimen is the oldest known chemidophorine (crown or not), period. The technique does not change what it is, only how it was evaluated”)—the technique does change what it is (see above), and I do not think that YPM VP 4607 is the oldest cnemidophorine from North America. My language here was completely intentional. As Reviewer 2 pointed out, there is an older fossil from the early Miocene of Florida that was referred to “Cnemidophorus” (Estes 1963). In my opinion, there is nothing about that fossil suggesting that it is not a cnemidophorine. However, in an apomorphy-based system, there is nothing about that fossil indicating that it is a *crown* cnemidophorine, based on the description and the illustration. That does not mean that the identification of the fossil as a cnemidophorine is erroneous. It means that there is currently not enough evidence to support an identification to the crown clade with an apomorphy-based diagnosis. The fossil could still be used to discuss the fossil record of cnemidophorines, but it should not be used to calibrate the minimum age of crown cnemidophorines, and I personally would be wary of using it at all in a divergence time analysis or in other types of analyses (biogeography, paleoecology, etc) until the identification is reevaluated.

To add to that last point, there are other fossils from the early Miocene of Florida (Chovanec 2014, a master's thesis cited here) that represent an older record of total clade cnemidophorines in North America. Thus, my paper describing the oldest known crown cnemidophorine. I look forward to the description of additional crown cnemidophorines, including specimens older than YPM 4707.

This is a technique that has been revised and upgraded significantly in the past 30 years. I suspect a new analysis of the deposit would return a different age.

Unfortunately, there is nothing else to go on. That age seems to be accepted by geologists and paleontologists who work on the Ogallala Group as far as I can tell, and mammal biochronology supports a somewhat older age for the fossil (see new paragraph in Material and Methods section).

We all want to have discovered "significant" specimens and I believe all of them are to one degree or another. In this context I believe the word "useful" is more appropriate.

Changed to useful, and agreed. This comment seems contradictory, given your comment below.

This paragraph represents the only referral to the paleobiogeographical importance of this specimen. I find this to be unfortunate. The emphasis in this manuscript is far too focused on the morphology and really misses the important contribution this specimen can make to testing hypotheses of the biogeographical history of this group of lizards...one of the key non-mammalian taxa of the Great Biotic Interchange. A more well-developed discussion of this would greatly improve the impact of this paper.

The fossil doesn't help test biogeographic hypotheses related to the Great Biotic Interchange, which occurred millions of years after the fossil was deposited.

I have added some more biogeographic discussion (new paragraph).

The figure is too washed out to discern detail properly. I was able to use the Levels tool in Photoshop to correct the image and reveal significantly more clarity. Images are data and must be of the highest quality possible.

Figures are too washed out to allow the subtle aspects of the anatomy to be clearly seen. I recommend using the Levels tool in Photoshop to reduce the white value (reduce the left side tail of the bright-dark curve).

Same for all specimen images.

I don't think the fossil images were washed out, but agree that modifications were needed for that figure and more so for the modern specimen figures (some of which I agree were washed out). All figures are now fixed.